# Observed interannual changes beneath Filchner-Ronne Ice Shelf linked to large-scale atmospheric circulation

Tore Hattermann[1,2,3 ✉], Keith W. Nicholls [4], Hartmut H. Hellmer [1], Peter E. D. Davis [4], Markus A. Janout [1], Svein Østerhus [5], Elisabeth Schlosser [6,7], Gerd Rohardt[1] & Torsten Kanzow [1,8]

Floating ice shelves are the Achilles' heel of the Antarctic Ice Sheet. They limit Antarctica's contribution to global sea level rise, yet they can be rapidly melted from beneath by a warming ocean. At Filchner-Ronne Ice Shelf, a decline in sea ice formation may increase basal melt rates and accelerate marine ice sheet mass loss within this century. However, the understanding of this tipping-point behavior largely relies on numerical models. Our new multi-annual observations from five hot-water drilled boreholes through Filchner-Ronne Ice Shelf show that since 2015 there has been an intensification of the density-driven ice shelf cavity-wide circulation in response to reinforced wind-driven sea ice formation in the Ronne polynya. Enhanced southerly winds over Ronne Ice Shelf coincide with westward displacements of the Amundsen Sea Low position, connecting the cavity circulation with changes in large-scale atmospheric circulation patterns as a new aspect of the atmosphere-ocean-ice shelf system.

[1] Alfred Wegener Institute Helmholtz Centre for Polar and Marine Research, Bremerhaven, Germany. [2] Akvaplan-niva AS, Tromsø, Norway. [3] Norwegian Polar Institute, Tromsø, Norway. [4] British Antarctic Survey BAS, Cambridge, UK. [5] NORCE Norwegian Research Centre and Bjerknes Centre for Climate Research, Bergen, Norway. [6] Department of Atmospheric and Cryospheric Sciences, University of Innsbruck, Innsbruck, Austria. [7] Austrian Polar Research Institute, Vienna, Austria. [8] Bremen University, Department of Physics and Electrical Engineering, Bremen, Germany. ✉email: tore.hattermann@npolar.no

Much of Antarctica is surrounded by floating ice shelves that buttress the flow of grounded ice into the ocean[1]. Over recent decades ongoing ocean-induced ice shelf thinning has driven increased mass loss from the Antarctic ice sheet[2]. Hence, understanding the fate of oceanic heat that flows toward the cavities beneath Antarctic ice shelves is one of the greatest challenges for determining future sea level. Located in the southern Weddell Sea, Filchner-Ronne Ice Shelf (FRIS, Fig. 1) is the planet's largest ice shelf by volume and is fed by the ice streams and tributaries that drain much of the marine-based West Antarctic Ice Sheet[3]. Unlike "warm cavity" ice shelves that are flushed by up to +1 °C waters of open ocean origin[4], most of the water entering the FRIS cavity remains close to the ocean's surface freezing temperature of −1.9 °C[5]. Consequently, loss of ice due to basal melting beneath FRIS is currently small considering the ice shelf's size[6]. However, observations indicate that FRIS is vulnerable to warmer inflows[7], and modeling studies suggest that FRIS may exhibit a tipping-point behavior, with the potential for an order-of-magnitude increase in basal melt rates[8,9] that might have profound impacts, within this century, on the ice sheet and dense water formation in this region[10].

Increased mass loss from FRIS can be triggered by a drop in the production of dense water on the continental shelf that currently prevents warmer offshore waters from entering the FRIS cavity[11]. Despite being in a phase of decline between 2006 and 2015[12], the prevailing cold and dry southerly winds that blow off the continent drive efficient sea ice formation in polynyas along the Ronne Ice Shelf front. The associated brine rejection densifies the continental shelf waters, forming High Salinity Shelf Water (HSSW) that drives a thermohaline circulation in the southward-deepening ice shelf cavity[5]. As a result of the pressure dependence of the melting point of ice, HSSW cooled to surface freezing temperatures can drive basal melting over the deepest parts of the ice shelf[13], creating a colder, but fresher, and therefore slightly less dense[14] water mass known as Ice Shelf Water (ISW). ISW is primarily exported from beneath FRIS through the Filchner Trough[15,16] that connects the ice shelf cavity with the deep ocean. At present, these outflows are dense enough to block the inflow of Warm Deep Water that circulates along the Weddell Sea continental shelf break[17]. For a warmer and wetter future climate, model simulations suggest that HSSW production is reduced far enough that the ISW density barrier will be eroded and warmer waters will be able to flush the FRIS cavity[11,18]. However, the uncertainties in these projections are large[19] and the likelihood of such a regime shift due to ongoing climate change remains largely unknown.

This study employs new observations of water masses, currents, and basal melt rates[20] beneath FRIS from five hot-water drilled boreholes (Methods). Together with ship-based observations from the ice shelf front[21,22], the data show that since 2015 a strengthening in wind-driven sea ice formation in the Ronne polynya caused an intensification of the density-driven circulation in the FRIS cavity. We observe dense water pulses propagating from Ronne Ice Shelf and driving a mode shift of the circulation under the northern Filchner Ice Shelf that affects the water mass properties in the Filchner Trough on interannual time scales. Combining our observations with analysis of satellite-based sea ice concentration[23,24] and atmospheric reanalysis products[25] further suggests that the sustained dense water production is related to recent atmospheric anomalies, in particular the position, and possibly the strength, of the Amundsen Sea Low (ASL) that occurred during an increasingly positive Southern Annual Mode (SAM). These findings further our understanding of how this sector of the marine-based West Antarctic Ice Sheet may respond to changes in large-scale climate and the implications for future sea level rise.

## Results

**Persistent cooling at Filchner Ice Shelf front since 2017.** The first 9–12 months of the ocean time series (starting in December 2016) from the northern drill sites on Filchner Ice Shelf (FNE1 & FNE2, Fig. 1a) show temperatures close to the surface freezing point (−1.9 °C, Fig. 2b), indicating the direct inflow of HSSW at least as far as 40 km inland of the calving front. While the warmest waters appear close to the seafloor, pulses of elevated temperatures between 600 and 800 m indicate inflows into the cavity at intermediate density/depth levels also.

The meltwater mixing-line relationship that describes how ocean temperature and salinity change through interaction with an ice shelf[14] indicates a local origin for the HSSW present in the Filchner Ice Shelf cavity in early 2017 (Fig. 3). During this period, observed source water salinities (Methods) between 34.84 g kg$^{-1}$ (34.65 psu) and 34.87 g kg$^{-1}$ (34.7 psu) at FNE1 and FNE2 (Fig. 2c) compare well with the Berkner HSSW[21,26] that is found on the shallow continental shelf between the Filchner and Ronne ice shelves north of Berkner Island[27].

The Berkner HSSW intrusions persist until July 2017, when the temperature of the entire water column begins to decrease gradually to below −2.25 °C in 2018 (Fig. 2b). This cooling indicates the arrival of more heavily glacially modified ISW. In contrast to the warmer Berkner HSSW intrusions, the colder ISW is associated with higher source water salinities[28] of around 34.92 g kg$^{-1}$ (34.75 psu) (Fig. 3). Similar source water salinities are observed at the southern Filchner drill sites (Figs. 1 and 2c; FSW & FSE) and relate to denser Ronne HSSW that is typically found in the southwestern Weddell Sea. This HSSW results from sea ice formation in polynyas in front of western Ronne Ice Shelf[21,27,29].

Although ISW outflow is a seasonal occurrence in the Filchner Trough[22], the observed cooling at FNE in 2017 marks a persistent change that governs the northern Filchner cavity for the remaining 18 months of the observational period. Conductivity temperature depth (CTD) transects in front of Filchner Ice Shelf (Fig. 2e, f) show how the Berkner HSSW that fills the lower 500 m of the Filchner Trough in 2017 was entirely replaced by Ronne HSSW-derived ISW in 2018. Based on this shift in source water mass properties, it appears that at different times one of two different circulation modes dominates under northern Filchner Ice Shelf (Fig. 1a): a Berkner mode where the Filchner Trough is filled with Berkner HSSW that enters the cavity at the Filchner ice front and determines the ISW properties under northern Filchner Ice Shelf, and a Ronne mode, where remotely sourced ISW, originating from HSSW inflows at the Ronne ice front, propagates through the FRIS cavity toward the northern Filchner Ice Shelf to dominate water mass properties in the Filchner Trough. While being most visible from changes in temperature and salinity, which contain information about the history of the water masses, the current meter records also confirm a shift from a largely partitioned cavity circulation where local inflows dominate the northern Filchner Ice Shelf (Berkner mode, green to purple arrows in Fig. 1a), toward a coherent circulation that spans the entire FRIS in response to increased HSSW inflow at the Ronne ice front (Ronne mode, orange to blue arrows in Fig. 1a), as discussed in further detail below and with a schematic of the associated overturning circulation being depicted in Fig. 11 in Janout et al.[21].

The ocean cooling beneath northern Filchner Ice Shelf during the transition into the Ronne mode is accompanied by a decrease in local basal melting as observed by downward-looking radars that were deployed near FNE1 in January 2016, and then at FNE1 from the time of the mooring installation (Methods, Fig. 2a). Considering the relatively strong tidal mixing beneath FRIS[30], and comparing with the pressure-dependent melting point temperature of ≈−2.25 °C beneath the 500–600 m deep ice base in this region,

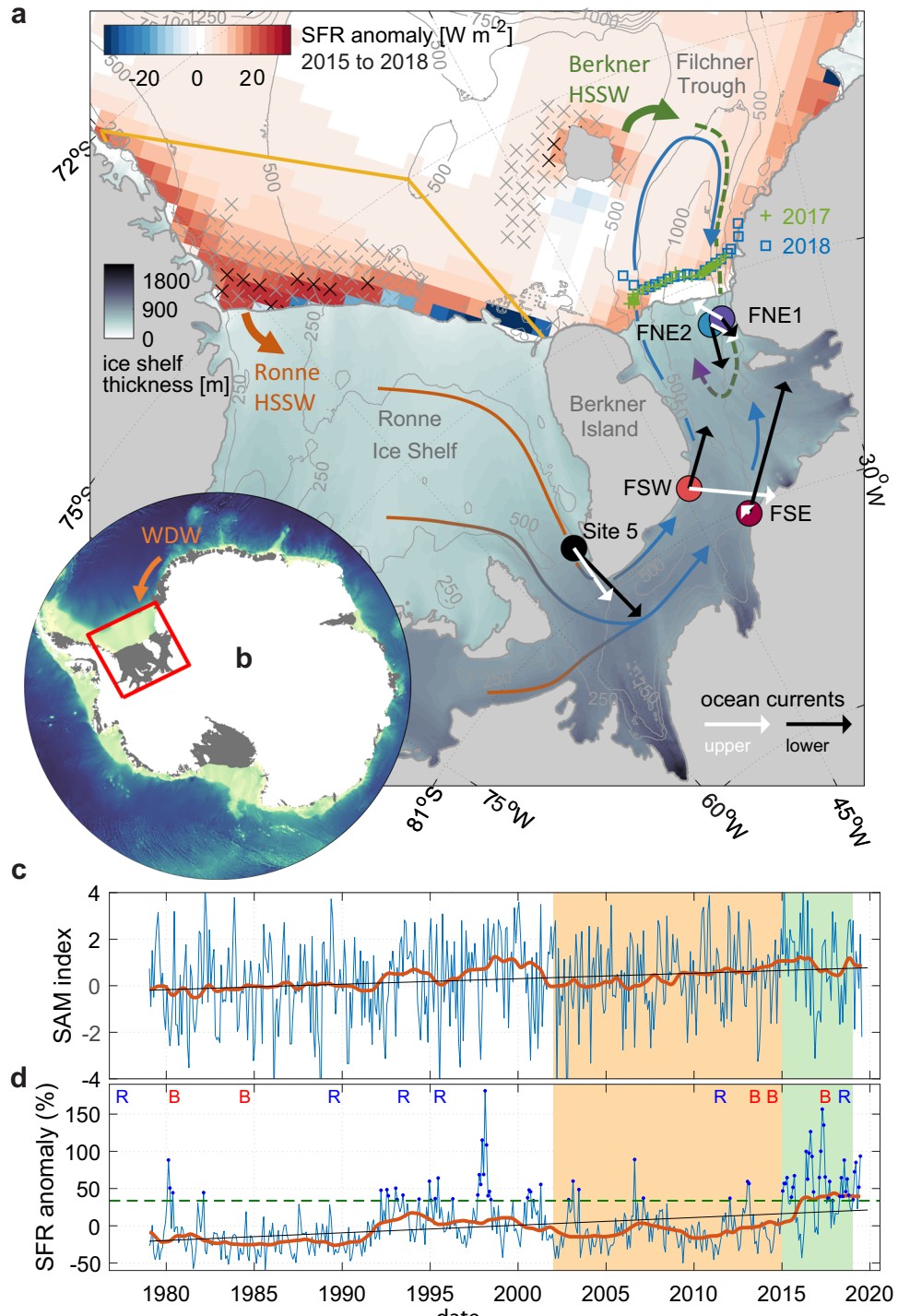

**Fig. 1 Overview of the Filchner-Ronne Ice Shelf system. a** Map of Filchner-Ronne hot-water drilled boreholes (Site 5, FSW, FSE, FNE1, & FNE2) and High Salinity Shelf Water (HSSW) production regions, (**b**) located south of the Warm Deep Water (WDW) that circulates along the continental slope in the southern Weddell Sea. **a** Contours: water column thickness (m); open ocean colors: anomalous sea ice formation rate (SFR) after 2015 compared with all years since 2002; blue (green) squares (crosses): ship-based CTD stations in front of the Filchner Ice Shelf in 2018 (2017); Vectors indicate time-averaged current meter observations from the lowest (black) and highest (white) instrument in the water column and legends showing a 2 cm s$^{-1}$ (10 cm s$^{-1}$) scale for the Filchner sites (Site 5). Colored arrows indicate inferred pathways and transformation from HSSW into ISW in the Berkner mode (green to purple) and in the Ronne mode (orange to blue). **c** Monthly and 36-month low-pass filtered time series of Southern Annual Mode (SAM) and (**d**) relative SFR anomalies, area-averaged over the orange polygon in front of Ronne Ice Shelf. Black lines: linear trend; blue dots: enhanced SFR events greater than 1$\sigma$ (green dashed line) above the time average; blue (red) letters: years of dominantly Ronne (Berkner) mode water masses in the Filchner Trough[21,22]. Black (grey) crosses in (**a**) indicate significant correlations between low-pass filtered time series of local SFR anomalies and SAM at 99% (95%) confidence level. Color shaded areas in (**c** & **d**) indicate the 15 year period (orange and green) relative to which the spatial pattern of SFR anomalies (green only) is computed that is shown in (**a**).

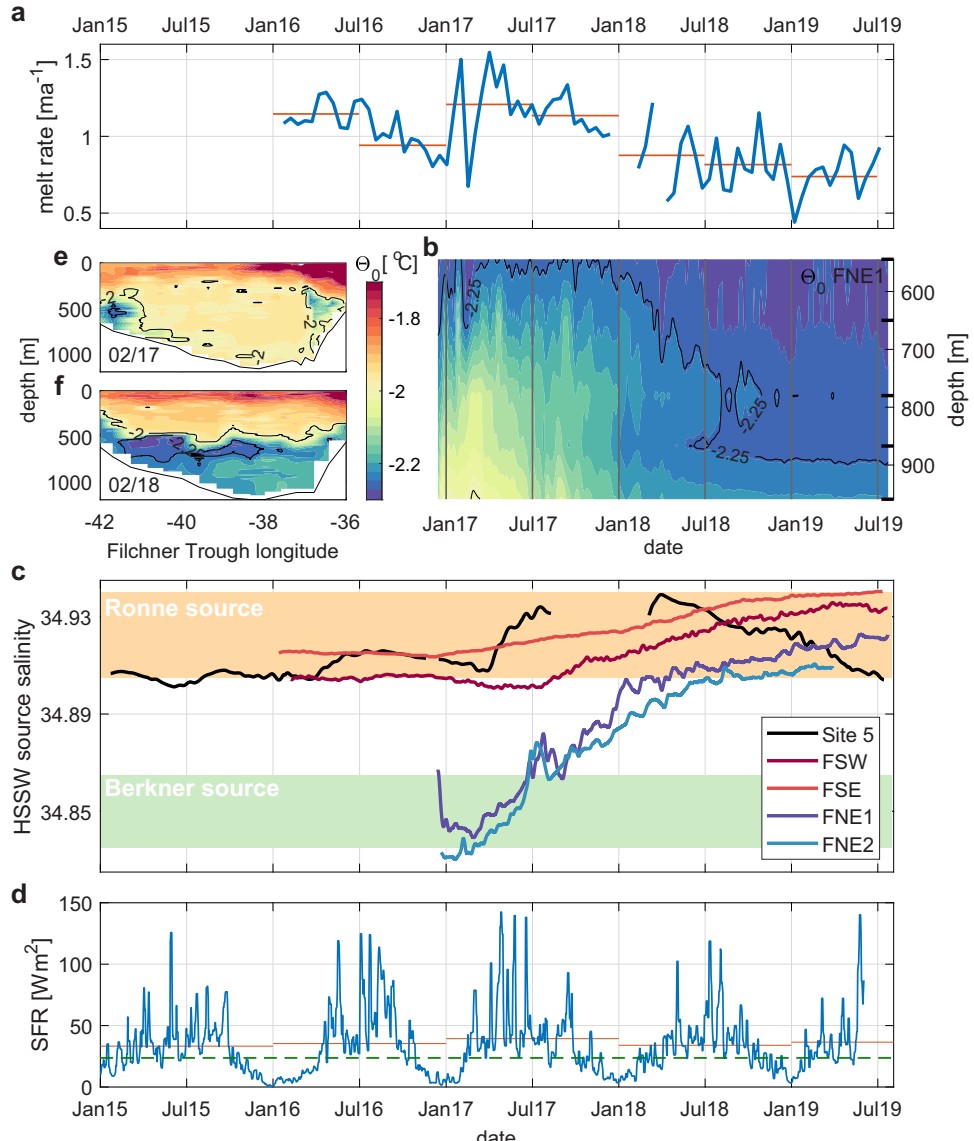

**Fig. 2 Indicators of a mode shift beneath Filchner Ice Shelf.** Time series of (**a**) ice shelf basal melt rates near FNE1, and (**b**) ocean conservative temperature at FNE1. **c** Source water salinities observed at each ice shelf mooring, indicating the Ronne (orange) and Berkner (green) HSSW regimes, and (**d**) 15-day low-pass filtered daily SFRs in front of Ronne Ice Shelf in comparison with the long-term average (green dashed line). **e**, **f** CTD sections of ocean conservative temperature at locations indicated in Fig. 1a along the Filchner Ice Shelf front for February 2017/2018. Thick black ticks along the y-axis in (**b**) indicate the depth of the temperature sensors at FNE1. Black −2.00 °C and −2.25 °C temperature contours in (**a**, **e**, & **f**) delineate the presence of ISW. Red horizontal lines in (**a**) and (**d**) show time-averaged values over shorter periods of each time series.

these relatively warm water intrusions provide heat to increase local melting at the ice base. In 2017, peak melt rates of over 1.5 meters per year consistently coincide with peak temperatures observed at the two uppermost instruments at FNE1, while melt rates drop during 2018 when temperatures stay below −2.2 °C at all sensors. The time-averaged melt rate for the period between January and July decreased from about 1.2 meters per year in 2017 to about 0.8 meters per year in 2019. In 2016, prior to the start of the ocean mooring time series, higher melt rates were also observed (Fig. 2a), suggesting that Berkner HSSW may already have been present inside the Filchner Ice Shelf cavity at that time. As melting deeper within the FRIS cavity may be expected to increase under enhanced inflow of Ronne HSSW that brings heat deep into the cavity[31], the mode shift that occurred in 2018 would indicate a shift in the balance of melting from the more northern Filchner Ice Shelf towards the grounding lines further south[32].

**Propagation of HSSW from Ronne to Filchner Ice Shelf.** Bulk estimates of sea ice formation rates (SFRs) in the southern Weddell Sea (Methods) reveal that extended periods of open water in front of western Ronne Ice Shelf have fostered anomalously high SFRs since 2015 (Fig. 1d) before Ronne HSSW-derived ISW arrived at the Filchner Ice Shelf front after 2017. A map of SFR anomalies (Fig. 1a) shows that heat loss in the Ronne polynya from 2015 to 2018 on average was more than 20 Wm$^{-2}$ higher than the average of the last 15 years (2003–2018, green and orange patch in Fig. 1c, d). The source water mass evolution at the different drill sites (Fig. 2c) shows the propagation of Ronne HSSW pulses through the ice shelf cavity during that time. Extended periods of enhanced sea ice formation in the austral winters of 2016 and 2017 (Fig. 2d) coincide with seasonally elevated source water salinities at Site 5 (black curve in Fig. 2c), which is consistent with a direct route for HSSW from the

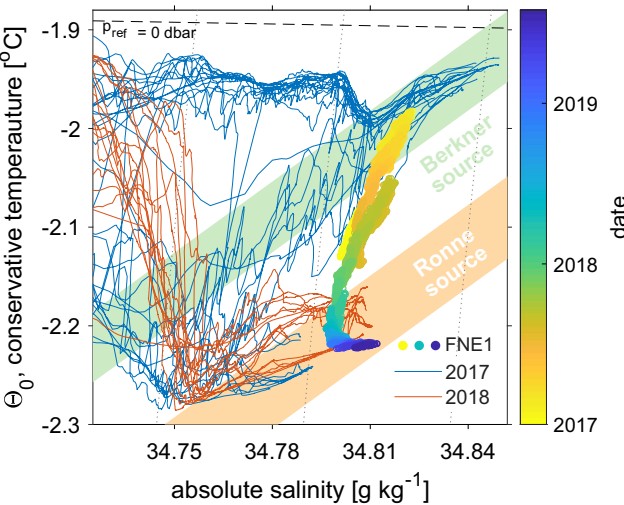

**Fig. 3 Conservative temperature–salinity diagram and HSSW source water types.** Comparing water mass time evolution at FNE1 (lowest sensor) with CTD observations at the Filchner Ice Shelf front (thin curves). Colored patches indicate the slope of the melt water mixing line[14] that relates a given ISW type to the temperature and salinity of its source. Together with the assumption that the source water is at the surface freezing temperature (black broken line) this is used to distinguish between the Ronne-sourced and Berkner-sourced water mass types in Fig. 2c. Thin black contours show potential density.

western Ronne Ice Shelf front toward the southern tip of Berkner Island[33] (Fig. 1a). In particular, the rapid source water salinity increase in early 2017 at Site 5 (black curve in Fig. 2c), followed by a similar signal at FSW later in the same year (dark red curve in Fig. 2c), stands out as leading the transition to the Ronne mode at the FNEs (blue curves in Fig. 2c).

The arrival of those Ronne HSSW pulses is also seen as a seasonal increase in low-pass filtered southward flow speeds that vary between 10 and 15 cm s$^{-1}$ at the upper and lower current meter deployed at Site 5 (Fig. 4a). Current meter measurements beneath Filchner Ice Shelf are generally dominated by strong tidal currents that can exceed 50 cm s$^{-1}$ during spring tides. This indicates a rather diffusive flow regime, in which precise transport estimates are challenging and the buoyancy-driven mean flow velocities only contribute as small residuals. However, while velocities at the upper current meters at the southern Filchner sites (Fig. 4c, d) are more influenced by local processes, the low-pass filtered data from the lower instruments show residual currents in the order of a few cm s$^{-1}$, nfirming the northward propagation of Ronne HSSW-sourced ISW along both flanks of the southern Filchner Ice Shelf (Fig. 1a). Seasonal temperature maxima occurring each year in October/November at FSW (Fig. 4c) suggest that Ronne HSSW pulses originating from mid-winter peak sea ice formation arrive with a 3–4 month lag east of Berkner Island. Considering the observed flow speeds between 10–15 cm s$^{-1}$ at Site 5 and less than 5 cm s$^{-1}$ at FSW, this is a plausible time scale for the HSSW signal to be advected along an approximately 800 km long pathway from Ronne Ice Front to FSW, although with some uncertainty about contributions from the tides.

Alongside the direct pathway revealed by the seasonal propagation of HSSW pulses from northern Ronne Ice Shelf via Site 5 along the western flank of the Filchner Ice Shelf cavity to FSW (Fig. 1a), the gradual increase in source water salinities at FSE on the eastern flank (light red curve in Fig. 2c) is witness to a less direct and slower pathway throughout the cavity. Source water salinities at FSE are generally higher than at FSW and start

to increase before the distinct pulse arrives at Site 5 in mid-2017. This suggests an influence of Ronne HSSW at FSE that has been cycled deeper inside the cavity, probably originating from a previous season, as is supported by tracer analyses that suggests an interannual propagation time scale throughout the FRIS cavity[21]. Such a delayed response to HSSW inflows along less direct pathways might also explain why source water salinities at all moorings under Filchner Ice Shelf continue to rise, when source water salinities at Site 5 gradually decline during 2018/19, likely due to fewer periods of high sea ice formation during 2018 compared with the previous 2 years (Fig. 2d). Ronne sourced ISW that eventually arrives at the Filchner Ice Shelf front hence represents the integrated signal from those different pathways.

Consistent with an intensified eastward propagation of saline Ronne HSSW through the FRIS cavity, potential density anomalies at FSW and FSE increase from below 32.68 kg m$^{-3}$ in 2016 to above 32.69 kg m$^{-3}$ in 2018, eventually causing a reversal of the density difference between the southern and northern Filchner drill sites (Fig. 4e). Together with an increase in northward velocities at FSE starting in 2018 (Fig. 4d), this highlights the interannual response of the cavity circulation as part of the shift into the Ronne mode, where outflows of dense Ronne HSSW-derived ISW dominate the circulation under the northern Filchner Ice Shelf, instead of the local inflows of Berkner HSSW that were observed in 2017.

Current meter observations at northern Filchner Ice Shelf (FNE1) show southward velocities at the lowermost instrument that prevail during the observational period regardless of the mode shift (Fig. 4b). During the Berkner mode, Berkner HSSW would be expected to enter the cavity along this pathway. During the Ronne mode, this supports the picture of a cyclonic circulation in the Filchner Trough (Fig. 1a), where Ronne HSSW-sourced ISW exits along the western flank[22] to recirculate and enter the cavity again along the eastern flank. The observed northward currents at FSE (Fig. 4d), however, contrast with earlier hypotheses that such a cyclonic circulation coherently extends all the way into the southern Filchner Trough[5]. Instead, our new mooring data indicate the existence of two, at least partially disjoint circulation cells beneath the northern and southern parts of the Filchner Ice Shelf, likely to be most pronounced during the Berkner mode (Fig. 1a). A recently discovered bathymetric feature running across the cavity may potentially be important for this partitioning of the cavity circulation[34], once more highlighting the need for accurate knowledge of the sub-ice shelf bathymetry.

More subtle changes in flow velocities are seen at the mid-depth and upper current meters at FNE1 (Fig. 4b), where northward (upper) and westward (mid-depth) velocities in 2017 become weaker (upper) and turn south (mid-depth), to approach a more uniform vertical structure in 2018. This is consistent with a reduction in the basal melting at the northern Filchner Ice Shelf that, during the Berkner mode, strengthens a local overturning circulation consisting of HSSW inflows in the lower layer and locally produced ISW outflows in the upper layer.

**Sea ice formation linked to recent atmospheric anomalies.** The multi-decadal time series of area-averaged SFRs in front of Ronne Ice Shelf (Fig. 1d) shows more frequent events of enhanced SFRs (>1σ above mean, dark blue dots in Fig. 1d) between 1992 and 2003, followed by a phase of decline after 2006[12], and an increase again from 2015. Some clusters of enhanced SFR events (Fig. 1d) occurred during periods of SAM in a stronger positive phase (Methods, Fig. 1c). In particular, the increase since 2015 that caused the reinforcement of the Ronne mode observed at our moorings, as well as the extreme maximum in 1998 that also affected the cavity circulation[35], coincide with periods when the

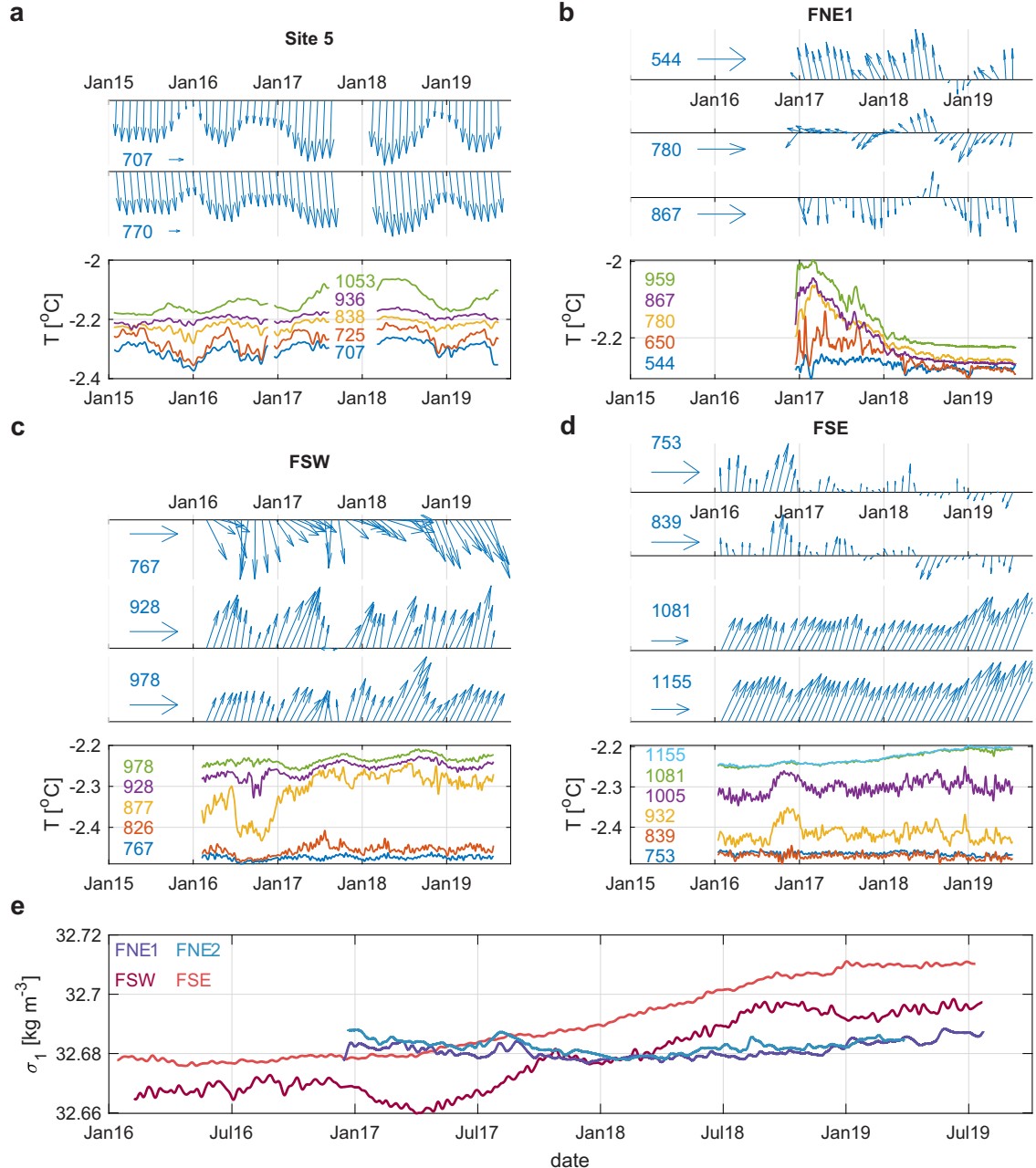

**Fig. 4 Overview of ice shelf mooring observations.** Time series of low-pass filtered velocity vectors (blue arrows) and in situ temperature (colored curves) at (**a**) Site 5, (**b**) FNE1, (**c**) FSW, and (**d**) FSE. Colored numbers indicate the instrument deployment pressure (dbar), horizontal arrows indicate an eastward current of 5 cm s$^{-1}$ as a reference scale for the respective axes. Time series (**e**) of potential density referenced to 1000 dbar from the lowermost sensors at the two northern and two southern drill sites on Filchner Ice Shelf showing the density reversal during the shift from Berkner mode to Ronne mode.

36-month low-pass filtered SAM index is more positive than a linear regression of the time series.

The SAM is the dominant mode of atmospheric variability in the extratropical Southern Hemisphere[36] and the SAM index has been generally increasing over recent decades. The linear fit to the area-averaged monthly SFR anomalies also yields an increase of $10 \pm 3\%$ ($4 \pm 3\%$) per decade between 1978 and 2019 (2015). Although correlations between SAM and SFRs are hardly significant on monthly time scales (Supplementary Table 1), the low-pass filtered time series share similar features and covary with a maximum at 11 months lag ($r = 0.70$, $r = 0.52$ after trend removal, Supplementary Fig. 1). Crosses in Fig. 1a furthermore indicate where SAM index and local time series of SFRs in the

southern Weddell Sea correlate at 99% (black) and 95% (grey) confidence.

Although this suggests a general connection between the activity of the Ronne polynya and large-scale climate indicators, the SAM explains only about one-third of atmospheric variability in the extratropical Southern Hemisphere and does not give any information about the longitudinal position of cyclones in the circumpolar trough. It is therefore likely that the SAM effect on sea ice production might be mostly a result of the lower air temperatures and higher wind speeds that would be related to a more positive SAM. Of greater importance for the atmospheric circulation is the longitudinal position of cyclones and anticyclones since this determines the direction of the atmospheric flow.

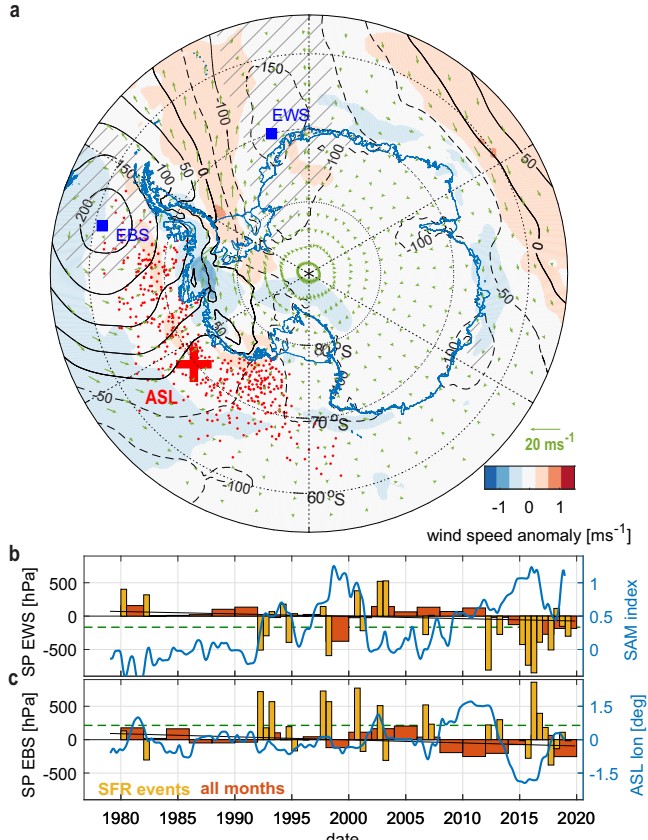

**Fig. 5 Atmospheric conditions associated with enhanced sea ice formation. a** Composite of atmospheric anomalies in surface pressure (SP in Pa, above 99% confidence level in hatched areas), wind vectors (arrows), and speed (color) associated with enhanced SFR events shown in Fig. 1d; red dots: monthly locations of the Amundsen Sea low since 1979 scattered around the time mean position indicated by the red cross. **b** Time series of SP anomalies located at the blue square in the eastern Weddell Sea (EWS), and (**c**) the Eastern Bellingshausen Sea (EBS), averaged for all months (only months associated with enhanced SFR events) in 2 years (6 months) bins. Green dashed lines indicate the average for all enhanced SFR events, black lines show the linear trend over all months. Blue curves show the 36-month low-pass filtered evolution of the large-scale indicators associated with favorable negative (positive) SP anomalies at EWS (EBS) that lead to enhanced southerly (offshore) winds at the Ronne Ice Shelf front. Negative ASL longitude anomalies indicate westward displacements that favor high pressure anomalies in the EBS.

To assess which atmospheric conditions may promote the Ronne mode beneath FRIS, monthly anomalies of surface pressure (SP), wind speed, and direction derived from ECMWF ERA-Interim[25] were averaged over all months with enhanced SFR events indicated in Fig. 1d. The resulting composite (Fig. 5a) shows that enhanced sea ice production is associated with stronger south-westerly (offshore) winds over Ronne Ice Shelf, which favor polynya formation north of the ice shelf front. Since air temperatures in this region are below freezing throughout the year[21], the amount of open water is the dominant factor for sea ice formation ($r = -0.70$ for monthly SFR anomalies and sea ice concentration, see Supplementary Table 2 for other variables). Note that to suppress self-correlations with atmospheric pressure fields in the composite average, SFRs were computed using a spatially varying, but time-averaged (1979–2019) wind speed, whereas an estimate using the time-varying wind speed (SFR*) is relatively strongly correlated with both the open water area and with the wind speed itself (see

Methods). However, the composite pattern remains unchanged for both definitions, SFR and SFR*, supporting robustness of the relation between the strength of the southerly winds in the southern Weddell Sea and activity of the Ronne polynya. The composite further indicates that the southerly winds are linked to synoptic-scale surface pressure anomalies that amplify the offshore katabatic winds over the Ronne Ice Shelf. Positive pressure anomalies associated with enhanced SFR events are centered above the eastern Bellingshausen Sea (EBS, Fig. 5a, c), whereas negative anomalies are seen in the eastern Weddell Sea (EWS, Fig. 5a, b), leading to an anticyclonic circulation with southerly flow over the FRIS front. Monthly time series of the difference in pressure anomaly between these locations significantly, albeit weakly, correlate at zero lag with the SFRs ($r = 0.33$).

The positive pressure anomaly west of the Antarctic Peninsula borders the Amundsen Sea Low (ASL), a climatological low pressure area in the Pacific sector of the Southern Ocean[37,38]. Variability in the depth and location of the ASL strongly affects climate and sea ice conditions in West Antarctica by influencing the meridional wind fields[37,39,40]. The position of the ASL is related to the existence (or non-existence) of an anticyclone in the EBS and thus the direction of the general flow across the ice shelf edge (Fig. 5 a). Monthly SP anomalies in the EBS covary significantly with anomalies of the zonal ASL position ($r = -0.52$, with an annual mean longitude of 232° E), indicating westward ASL movements as a factor promoting enhanced sea ice formation and hence dense water production in the southern Weddell Sea. Indeed, the recent transition from a phase of decline in polynya activity[12] to enhanced SFRs that reinforced the Ronne mode beneath FRIS coincides with a shift from a more eastward to a more westward ASL position (Fig. 5c). Negative pressure anomalies in the EWS are an additional factor that strengthens the offshore winds over the Ronne Ice Shelf. During the strong sea ice formation event in 2016, both factors were present (yellow bars in Fig. 5b & c) together with a westward displaced ASL and a strong positive SAM index (blue curves in Fig. 5b, c).

While positive SP anomalies in the EBS during enhanced SFR events appear largely independent of the long-term trend, the frequent occurrence of extreme low pressure events in the EWS since 2015 aligns with an overall decreasing SP anomaly as part of the more positive SAM ($r = -0.69$). The increased ice transport northward from the Ronne Ice Front is supported by an increased sea ice export from the northern Weddell Sea due to recently stronger westerly winds under a more positive SAM. Note that the ice drift deviates slightly to the left of the main wind direction[41], which is of minor importance for the clockwise transport around cyclones, but in the case of strong westerly winds leads to a northward displacement of the ice (edge), which releases pressure from the ice pack further south. In particular, this was the case in austral spring 2016/2017, which exhibited an extraordinarily early start of the melt season, and with positive anomalies in sea ice extent in the Weddell Sea related to atmospheric forcing[42,43]. This mechanism would be consistent with the apparent delayed increase of SFRs one season after a more positive SAM. Also, the Maud Rise polynya, which has earlier been linked to wind-driven upwelling was more pronounced during that period[44], as well as exceptionally warm and prolonged inflows of Warm Deep Water that were observed on the eastern continental shelf in the southern Weddell Sea[45]. Although the exact mechanisms controlling each of these anomalies remain to be resolved, their coincidence may point at a common origin, such as general intensification of the circulation associated with a strong positive SAM that persisted over this period.

## Discussion

Our new multi-annual oceanographic time series beneath Filchner-Ronne Ice Shelf show that recently intensified sea ice formation in the southern Weddell Sea has reinforced the cavity-wide buoyancy-driven circulation (Ronne mode) that maintains the dense water outflow in the Filchner Trough. This transition from a more localized overturning circulation under the northern Filchner Ice Shelf (Berkner mode) was caused by pulses of Ronne HSSW that propagated through the cavity along different pathways. Herein, the ice shelf acts as a delayed low-pass filter, such that the variability of HSSW inflows from the Ronne polynya affects water mass properties in the Filchner Trough on interannual time scales.

Our analysis further suggests a predominantly wind-driven control on SFRs in front of Ronne Ice Shelf in the present climatic regime. The anticyclonic atmospheric circulation that intensifies sea ice formation in the southern Weddell Sea is driven by synoptic-scale pressure anomalies that are linked to variability of the Amundsen Sea Low and the Southern Annual Mode. While this differs from the suggested gradual decline in dense water production that would make FRIS more susceptible to warm inflows in response to future climate change[11], further work is required to find out whether the reinforcement of the Ronne mode is a persistent trend or whether it stems from the internal variability of the system.

HSSW formation is also influenced by glacial meltwater inflow[46]. A sustained Berkner mode would eventually decrease the ISW density on the eastern part of the continental shelf, causing a potential feedback on the Berkner HSSW production and weakening the density barrier[11] that prevents modified WDW from entering the ice shelf cavity[7,18]. Although varying amounts of remotely sourced Ronne ISW and more locally produced Berkner HSSW have been observed in front of Filchner Ice Shelf (as indicated by blue and red letters in Fig. 1d), the density in the lower Filchner Trough has remained remarkably stable throughout recent decades[21,47]. This suggests that such feedbacks indeed exist, where the outflow of Ronne HSSW-sourced ISW acts as a precursor for the Berkner HSSW production (i.e., by determining the density at which convection that regulates HSSW production may occur). However, most of the meltwater from FRIS is currently exported through the Filchner Trough and does not affect the western continental shelf regions where the densest HSSW is produced that drives the Ronne mode. This indicates that the polynya activity in front of Ronne Ice Shelf really is the main regulator of the circulation of the southern Weddell Sea continental shelf.

Our study emphasizes the link between the atmospheric forcing and the oceanic circulation, which is shown by our measurements on an interannual-scale. Long-term implications of the described physical mechanisms depend on changes in the mean atmospheric conditions above the Southern Ocean. The majority of climate models exhibit an increasing SAM index in the 21st century, associated with a deepening and change in location of the ASL[37,48,49]. It depends, however, on the exact response, whether this should support stronger southerly winds over the Ronne Ice Shelf and an increase in the area of open water that facilitates increased dense water production at the ice shelf front. Changes in ASL location and strength were previously employed to explain the interannually varying warm water inflows toward ice shelves in the Amundsen Sea[50,51]. Our results imply a potential see-saw behavior between basal melt rates in that region and dense water production in the southern Weddell Sea on interannual time scales. However, the complex relationship between the ASL and hemispheric modes of climate variability (e.g., ENSO[52,53]) paired with the extremely high interannual variability of SP in the Amundsen-Bellingshausen and Ross Seas

regions, hampers accurate predictions of the longitudinal position and strength of the ASL in a changing climate. In particular, the Antarctic Dipole[54–56] connects tropical signals to Antarctic sea ice, where the Atlantic and Pacific centers of the dipole react in opposite ways to temperature anomalies in the tropics. Warm ENSO events are related to higher (lower) temperatures and less (more) sea ice than on average in the Pacific (Atlantic) and a persisting high-pressure center in the Bellingshausen Sea, the response usually lagging the tropical signal by several seasons. A very strong warm ENSO event in 2015/16 (defined using the ONI (NOAA)) might be connected to the increase in SFR after 2016 found in this study. Those ENSO-induced anomalies in Antarctica are modulated by SAM depending on their phase relationship[57,58]. McKee et al. found[54] statistically significant relationships between Weddell Sea Bottom Water temperature and ENSO/SAM. They suggest that dense shelf water production might be influenced by anomalous winds via modulating the fraction of open water over the shelf. Our study proves this hypothesis and similar mechanisms were found for the Ross Sea in the Pacific sector of the Southern Ocean[59], which is under more direct influence of ENSO.

Although the statistical evidence from short available time series is weak, our results highlight a new aspect of the atmosphere-ocean-ice shelf system[60,61] that deserves further attention. Reduced sea ice formation rates in a warmer, wetter climate have been suggested as the future paradigm for the southern Weddell Sea[11]. The mechanisms revealed by our study imply potentially wide ranging teleconnections and lend atmospheric projections predictive power over ocean conditions beneath an ice shelf. That is, if we know how the remote atmospheric patterns are likely to change (through coupled climate models), we can improve predictions about the future sea ice formation rates that will dominate the long-term response at the Filchner-Ronne Ice Shelf.

## Methods

**Ice shelf mooring data and hydrography.** Three consecutive hot-water drilling campaigns were conducted as part of a joint field program led by the British Antarctic Survey and the Alfred Wegener Institute, to deploy oceanographic moorings beneath the eastern Ronne Ice Shelf in 2014/15 at the previously surveyed Site 5[33]; in 2015/16 at a southeastern site (FSE) and a southwestern site (FSW) beneath the Filchner Ice Shelf; and in 2016/17 at two northern sites (FNE1 & FNE2) beneath the Filchner Ice Shelf (Fig. 1a). All moorings were equipped with 2–4 Nortek Aquadopp acoustic current meters and 2–6 Seabird SBE37 conductivity-temperature-depth (CTD) sensors, as well as RBR CTD sensors at Site 5. The different instruments are roughly spaced evenly between the ice shelf base and the seafloor at each location. The instruments are sampling at 2-hourly intervals, and carry internal batteries designed to last for a minimum of 5 years. Data are transferred via an inductive cable loop to an autonomous, solar powered data logger at the surface of the ice shelf, from where it is sent off the Antarctic continent via Iridium short burst data (SBD) messages for further analysis. Each site was re-visited for service and data download 1 year after deployment and a new surface logger was installed at FNE2 in austral summer 2017/18. The Iridium link was generally reliable, but the quality of data retrieved varied across the sites, such that physically implausible data were discarded during post-processing of the SBD messages.

Recovery and post-calibration of the mooring instruments are not possible. Instead, an SBE 49 fast-cat CTD was used to collect between two (FSW) and six (FSE) calibration profiles through the sub-ice shelf water column at each site prior to the mooring deployment. To minimize icing in the conductivity cell, the instrument was stored in a heated bath at the surface of the ice shelf, but sometimes needed to flush for a few minutes below the ice shelf base until conductivity readings had stabilized. CTD profiles were corrected after post-calibration of the instrument and the raw data were smoothed, corrected for thermal inertia of the cell and averaged into 1 m vertical bins using the SeaBird processing software. Absolute salinity and conservative temperature were computed using the TEOS-10 Gibbs thermodynamic potential for seawater. For comparability with earlier studies, source water salinities are also stated in units of practical salinity, following the Unesco 1983 algorithms for computation of fundamental properties of seawater[28]. Constant temperature and salinity offsets were subtracted from the moored data by comparing the first 10 measurements of each mooring sensor after its deployment with the post-deployment calibrated borehole CTD profile.

Subsequent drift in each instrument was ruled out by cross-validation of trends between neighboring instruments.

All CTD data were interpolated on a common 2-hourly time vector and then smoothed with a 27-day Hanning window moving-average filter using the jlab MATLAB-toolbox. Time series from temperature sensors of all five instrument-depths at FNE1 are shown in Fig. 2b, all other CTD data shown in this study are taken from the lowermost instrument at each site, typically being located a few tens of meters above the seafloor. Current meter velocities were rotated to correct for magnetic declination using the International Geomagnetic Reference Field (IGRF) Model MATLAB-toolbox, averaged in 5 day bins and smoothed with a 60-day low-pass filter. In Fig. 4a–d time series vectors of every 6th (30 days) smoothed bin average is shown for sensors at all depths per site. In Fig. 1a the time average over the entire time series is shown for the lowermost and uppermost (located few meters below the ice base) instrument at each site.

Ship-based CTD profiles along the Filchner Ice Shelf front (Fig. 1a) collected in January 2018 during cruise PS111 with RV Polarstern[21] and in January 2017 during cruise JR16004 with RRS James Clark Ross[22], are used to provide context for the sub-ice shelf measurements (Fig. 2e, f) and to assess the variability of water masses in the Filchner Trough in earlier years (Fig. 3).

**Source water salinity derivation**. When ice melts into seawater, the resulting water mass will cool and freshen at a constant ratio[14], causing water mass transformation along characteristic lines in temperature–salinity (T–S) space. Neglecting minor corrections due to heat conduction at the ice–ocean interface, the slope of these lines is given as $\partial T/\partial S = L/(S_0 c_p)$, with specific heat capacity of seawater $c_p = 4186\,\mathrm{J\,kg^{-1}K^{-1}}$ and latent heat of fusion $L = 3.34\times 10^5\,\mathrm{J\,kg^{-1}}$ and salinity $S_0$ of the seawater that is being in contact with the ice. In particular, in absence of other freshwater sources or T–S end-members, Gade-line theory can be used to invert for the open ocean water mass type that has been the source water for a given ISW type, as e.g., done by[26] to discriminate between Berkner HSSW and Ronne HSSW source water types for FRIS ISW. Assuming that ISW being observed at the mooring sites originated from HSSW on the continental shelf, the "source water salinity" of this HSSW is found as the intersection of the ISW's Gade-line with the surface freezing point temperature. Based on this, time series of source water salinity was computed from the filtered in-situ temperature and salinity data from the lowermost CTD instrument at each ice shelf mooring (Fig. 2c) that allow to compare the origins of water masses that have experienced different degree of modification due to their interaction with the ice shelf.

**Sea ice formation rates**. To assess the evolution of HSSW production on the southern Weddell Sea continental shelf, daily averages of open-ocean heat loss were calculated as a proxy for sea ice formation rates (SFRs), using passive microwave sea ice concentration (SIC) from the Special Sensor Microwave Imager/Sounder (SSMIS)[23,24] and atmospheric reanalysis data from ERA-Interim[25]. A bulk formula $\mathrm{SFR} = \rho^a c_p^a c_s (1 - \mathrm{SIC}) U \Delta T$ was used with constant parameters for density $\rho^a = 1.3\,\mathrm{kg\,m^{-3}}$ and specific heat capacity $c_p^a = 1000\,\mathrm{J\,kg^{-1}\,K^{-1}}$ of air and a sensible heat transfer coefficient $c_s = 1.5\times 10^{-3}$, together with model output for 10-m absolute wind speed ($U$) and the difference between the 2 m air temperature ($T$) and the ocean surface freezing temperature, computed as $\Delta T = -1.9\,°C - T$. Time series of 0.125 degree spherical grid resolution atmospheric fields were interpolated onto the 25 km x 25 km polarstereographic SIC grid. To avoid edge effects near the time-evolving ice front, only SIC grid points that are masked as ocean points during the entire time series are being used. Despite their relatively low spatial resolution, these products provide long time series, and comparison of the obtained SFRs with estimates from shorter higher resolving sea ice concentration products, as well as sea ice assimilating model simulations[21] yield similar results for the overlapping time periods, confirming the robustness of the method.

Spatially varying fields of absolute SFR anomalies averaged from 2015 to 2018, relative to the period from 2002 to 2018, are shown in Fig. 1a. Area-averaged time series of SFR, SIC, $\Delta T$ and $U$ were computed as the arithmetic average of all grid points within the polygon at the Ronne Ice Shelf front shown in Fig. 1a. High-frequency area-averaged absolute daily SFRs, smoothed with a 5-day moving median filter, are shown in Fig. 2d. For all other purposes, low-frequency time series were computed by averaging daily values into monthly bins. Monthly anomalies shown in Fig. 1c, and used in subsequent correlation analysis, were computed by removing the seasonal variation, i.e., subtracting the long-term average of all Januaries, Februaries, etc. from the respective monthly value of the time series, and scaling absolute values relative to the long-term average of all months. Low-pass filtered time series in Figs. 1, 4 and Supplementary Fig. 2 were computed using a 36-month moving median filter and subsequent spike removal with a 7-month Hanning window moving-average.

Correlations between monthly anomalies in sea ice formation, SIC, $\Delta T$ and $U$ are shown in Supplementary Table 2. To distinguish the dynamic control of winds on SIC from their direct influence in the bulk formula, and to suppress self-correlations with atmospheric pressure fields, monthly averaged SFRs were computed using a spatially varying, but time-averaged (1979–2019) wind speed. Area-averaged time series of sea ice formation rates using a time- and spatially varying wind speed in the bulk formula (SFR*) are shown in Supplementary Fig. 2,

for comparison. This because the bulk formulation is basically a linear combination of wind velocity, temperature, and sea ice concentration. Hence, if SFR* would be used to select months of the enhanced sea ice formation events for the atmospheric composite analysis, then months with higher wind velocities would a priori be preferably chosen. By using the SFR estimate based on time-averaged winds, this autocorrelation is suppressed and the composite provides more robust evidence of a relationship between reduced sea ice concentrations and enhanced offshore winds at the ice shelf front, as two independent variables.

**Climatic indices, atmospheric composite, and significance**. Large-scale climate variability is assessed using monthly time series of the observation-based Southern Hemisphere Annular Mode Index (SAM index)[36] downloaded from https://legacy. bas.ac.uk/met/gjma/sam.html (accessed April 3, 2020) as well as a reanalysis product of Amundsen Sea low (ASL) indices[49], version 2 downloaded from https:// scotthosking.com/asl_index (accessed April 3, 2020). Atmospheric composites and time series of seasonally corrected pressure anomalies were computed using monthly averaged 0.125 degree ERA-Interim surface pressure (SP) and 10-m wind velocity vectors and speed fields. Time series of local seasonally corrected SP anomalies in the eastern Weddell Sea (EWS) and the eastern Bellingshausen Sea (EBS) were calculated from spatial averages of all grid points within a radius of 50 km of the locations indicated in Fig. 5.

Unless stated otherwise, correlation coefficients cited in the main text and summarized in Supplementary Tables 1 & 2 were calculated using unfiltered monthly time series at zero lag, with significance being evaluated for $p = 0.01$. Covariance was computed for monthly and 36-month low-pass filtered time series of SFRs and SAM index with lag between 0 and ±18 months (Supplementary Fig. 2). Significance of correlations of the low-pass filtered time series was evaluated using a Monte Carlo bootstrapping method that appears more powerful for discarding insignificant correlations between the highly self-correlated series than the $p$ value testing. In this method, the correlation of a given SFR anomaly with SAM index is compared to probability distributions of correlations obtained from 1000 surrogate time series of randomly permuted and subsequently low-pass filtered SAM index-values. The same method is also used to assess the significance of the atmospheric pressure anomalies associated with enhanced SFR events shown in the composite Fig. 5a.

Note that different atmospheric reanalysis products may exhibit varying quality of agreement with measurements from meteorological stations, and varying decadal trends in wind speed and SLP. In our study, the atmospheric reanalysis data are foremost used to identify large-scale atmospheric patterns that favor wind-driven sea ice formation in front of the Ronne Ice Shelf. As discussed in the previous section, those occasions are primarily identified through anomalous low sea ice concentration. Hence, the selection of months to be included in the composite average of atmospheric anomalies will primarily be determined by the sea ice product and it is reasonable to assume that most reanalysis products will capture similar large-scale circulation anomalies during those times, despite the differences they may display under a more detailed comparison.

**Basal melt rates from downward-looking phase-sensitive radar**. Time series of basal melt was obtained from two Autonomous phase-sensitive Radio Echo Sounders (ApRES), one of which was deployed approximately 6.3 km west of FNE1, the other being deployed at FNE1 itself (see Fig. 1, main text). Time series from both instruments overlapped in 2017 and the combined time series covered the period from January 2016 to July 2019.

ApRES uses the frequency-modulated continuous wave approach, transmitting a tone that scans from 200 to 400 MHz over a period of 1 s, at an output power of 100 mW. The instrument measures the change in range from the antennas of the radar to the ice base, and, after appropriate processing, the data can yield a time series of basal melt rates. Once every 2 hours the instruments collected a burst of 25 measurements over a period of about 26 s. To process the data, each burst was averaged, and then its Fourier transform calculated using the methodology described by[20]. The result is a sequence of radar returns, which retain both the phase and amplitude of the signal.

Each return shows a strong reflecting horizon at a depth of about 610 m, indicating the depth of the ice base below the surface, assuming a dielectric permittivity of 3.18 throughout the ice column. The accurate depth of the ice base is not required here, but the phase sensitivity of the measurement means that the vertical motion of the base with respect to the radar antennas can be monitored with a precision that formally depends on the signal-to-noise ratio of the signal: a high signal to noise ratio of 60 dB, as in the case of the bed reflection in the present study, yields a range precision of less than $10^{-7}$ m. However, this thickness variation is the result of a combination several effects: basal melting, strain thinning in the ice column, vertical compaction in the firn, sinking of the antennas in the snow, and the temperature sensitivity of the instrument itself. The accuracy of the melt rate estimates is determined by the ability to account for these other effects.

By assuming that internal reflecting horizons from within the ice column are fixed in the ice, we use their changes in range to determine the non-melt-induced contributions to the thickness change, including the apparent contribution from the temperature-induced variations in the instrument. In principle, this allows us to find the thickness change contribution due to basal melting. However, if the internal reflectors are not flat and continuous, or are off-nadir, the vertical strain is

contaminated by the horizontal strain component, and unambiguously removing its effects is more difficult.

In addition, the strength of internal reflections is relatively low, their phase is known with less precision, and short-term (diurnal and faster), non-melt-induced variations in thickness are difficult to extract. This is important for tidally induced vertical strain, which is expected to make a strong contribution in this region. For the purposes of this study, in which longer-term variability is of interest, this difficulty can be simply overcome by filtering the signal to remove variability at time scales of the dominant tidal frequencies.

Here, 5-day bin averages were computed from the hourly data (and including data from both instruments in the overlapping period in 2017) to obtain basal melt rate time series (Fig. 2a) uncontaminated by other factors affecting the ice shelf thickness. As a result of the effects discussed above, the estimated error of the derived melt rate is 0.1 m a$^{-1}$.

## Data availability

Processed and filtered ice shelf cavity mooring and melt rate data presented in this study are available at https://doi.org/10.1594/PANGAEA.928758, with raw data available from the authors upon request. CTD data from PS111 are available at https://doi.org/10.1594/PANGAEA.897280, CTD data from JR16004 are available at https://doi.org/10.17882/54012. Sea ice data are available from NOAA/NSIDC Climate Data Record of Passive Microwave Sea Ice Concentration at https://doi.org/10.7265/N55M63M1 and https://doi.org/10.5067/U8C09DWVX9LM. ERA-Interim reanalysis data are available at http://apps.ecmwf.int/datasets/data/interim-full-daily/levtype=sfc/. The station-based SAM index is available at https://legacy.bas.ac.uk/met/gjma/sam.html (accessed May 3, 2021), the Amundsen Sea Low indices are available at https://scotthosking.com/asl_index (accessed May 3, 2021).

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

## Acknowledgements
T.H. is grateful to E. Darelius and R. Graversen for discussions and suggestions during the development of this study. T.H. was funded by the Alfred Wegener Institute strategic grant and the Norwegian Research Council grant 229764. K.W.N. and P.E.D.D. were funded by the UK Natural Environment Research Council under grant NE/L013770/1. SØ was funded by the Norwegian Research Council grants 229764 and 247699 and the EU Horizon 2020 grants 820575 and 821001.

## Author contributions

H.H.H., K.W.N., S.Ø., and T.H. conceptualized the study. K.W.N., T.H., P.E.D.D., S.Ø., H.H.H., and M.A.J. conducted the investigation. H.H.H., K.W.N., T.K., and S.Ø. acquired the funding and provided supervision. T.H. led the formal analysis and wrote the original draft. E.S. contributed validation, G.R. contributed data curation, all authors contributed to the writing by reviewing and editing the manuscript.

## Funding

## Competing interests

The authors declare no competing interests.
