## [Peer Review File · Nature Communications]

REVIEWER COMMENTS

Reviewer #1 (Remarks to the Author):

Main Comments

The manuscript by Hattermann et al is exciting and shows results that strongly improve our understanding of the Antarctic system. They show how remote atmospheric forcing (SAM and Amundsen Sea Low) regulates sea ice formation in the southern Weddell Sea, and consequently, basal melt of the largest ice shelf on the planet (Filchner Ronne Ice Shelf, FRIS). The new observations collected by the authors from beneath FRIS are unique and are the only ones that allow direct connections between remote forcing and ice shelf basal melt. The analysis is also accurate. The manuscript is relevant to the broader climate science community and therefore, I believe, appropriate for Nature Communications. I only have a couple of main comments that I believe need to be addressed for the manuscript to be consistent with what these new data show and with previous knowledge.

1) The overall message of the manuscript is a bit different compared to what the observations show. When I first read title and abstract I thought the manuscript would have shown "decadal to centennial results", similar to Hellmer et al 2012. Instead, the new data, although incredibly important, are "only" few years long and making strong inferences and connections with processes that occur over much longer time scales (e.g. warming of the southern Weddell Sea) is probably too "audacious". I would redirect the overall message (title and abstract) toward interannual variability and leave maybe one paragraph in the discussion to talk about long term implications. Interannual variability is key here and it has implication for basal melt and dense water production. Changes in these processes can have global consequences. Very important is the remote forcing on these interannual processes, as you describe.

2) Some important links to previous knowledge are missing in the present manuscript. First, Mckee et al 2012 (JGR, and other studies) show that changes in sea ice formation in the southern Weddell Sea can be associated with ENSO/SAM and this is not clearly expressed here. Second, there is a wide literature on the Antarctic Dipole (opposing sea level pressure in the Bellingshausen and Weddell seas) starting from Yuan and Martinson 2001 (GRL). Given the results shown here are associated with the Antarctic Dipole, a reference (and discussion) to these studies is important. Third, increased sea ice formation has been argued to actually increase basal melt in cold ice shelf cavities (e.g. Nicholls 1997, Nature) because more "HSSW flushing" brings more heat into the cavity. This is the opposite of the message of this manuscript, and therefore a more detailed comparison with these previous results is required. Finally, HSSW formation is also influenced by glacial meltwater inflow (e.g. Silvano et al., 2018 Science Advances, Hellmer et al 2017, JCLIM etc.) and this process is not discussed here despite it seems relevant for the overall story.

I would introduce/discuss this previous knowledge in the introduction/discussion and put strong emphasis on the amazing new data shown here (from the ice shelf cavity). Only these new data provide definitive evidence of the remote connection between the atmosphere and the "dark" ice shelf cavities.

Minor comments

- Figure 1: Specify in the caption what the "orange and green periods" are in panel C and D. Also mention what "upper" and "lower" ocean currents mean.
- Line 145-147: I would refer to Fig. 2A here to highlight the high melt rate in 2016.
- Line 147-150: this sentence is a bit unclear, or at least not explained. Please provide some further explanation of why more HSSW can cause increased basal melt deeper in the cavity. This also opposes the main message of the manuscript (more HSSW less melting). Moreover, in terms of ice sheet

stability, the deeper portions of the cavity are actually more relevant. See also the main comments.

- Line 152-188: It might be a good idea to include in a map the circulation pathways established in this study.
- Line 201: Maybe this is Fig S2.
- Line 211: "Large scale climate indicators".
- Line 242: sea ice moves roughly aligned with the winds (e.g. Holland and Kwok, 2012, Nat. Geosci.). So the Ekman forcing might not be relevant here.
- Line 261: Rapid changes have been observed in the western peninsula, while on the eastern peninsula (closer to FRIS) changes are less "strong" (except for ice shelf disintegration).

Reviewer #2 (Remarks to the Author):

General Comments

The paper presents new observations from the Weddell Sea and the Filchner-Ronne Ice Shelf cavity and an analysis of the observations. The results are very interesting not only for the community focused on FRIS, but also for the wider ice shelf-ocean community. I found the analysis of diverse observations clever and fascinating. I think the paper will be a great addition to the knowledge about ice shelf cavities after appropriate corrections are made.

The paper describes the following process. Recent changes in the sea ice production, caused by atmospheric anomalies, lead to a new circulation pattern in the FRIS cavity. As a result of this circulation, Filchner Ice Shelf frontal region experiences outflow of Ice Shelf Water, traced back to the Ronne polynya, and not ISW produced locally. This outflow changes the structure of the water column in the Filchner Trough.

I found the title leaning towards "clickbait". Although I can see it being very attractive, I feel that the paper did not match the expectations set by the title and I suggest changing the title. Two main concerns arise here. First, your analysis is nuanced and well-constrained, and suggests that no long-term effects on warm water inflow can be predicted. Thus "reduce threat" feels like an oversell to me. Second, I feel like the paper did not talk about warm water enough to be reflected in the title. You describe a new circulation pattern in the cavity, and contrast it to one modelling study (reference 13). I think it is very important to talk about implications of the new circulation pattern, as you do in the last section on the paper, but this is not the main result of your paper.

To justify the title, I think one has to present some evidence of warm waters inflow in the Filchner Trough, and the reversal of the signal in recent years.

Additionally, while you mention that FRIS is the largest ice shelf by volume (line 39), the title "largest ice shelf" is typically used for the Ross Ice Shelf.

In general, the paper reads as rushed, and many small inconsistencies make it hard to appreciate the science. A lot of work has to be put into reading the paper and deciphering the relationship between text and figures. In my opinion, the authors did not make the material easy to understand.

Major Comments

1. I am confused about your choice to talk about 2 modes of circulation, and not discuss observed velocities more (first at lines 115-122, and elsewhere). Did water flow direction change at FSs during 2017? Did water flow direction change at Site 5 (possibly twice, in 2017 and 2018)? Having multiple velocity measurements throughout the water column, could you see the difference in how top and bottom layers reacted? Is cooling in the bottom of FNE1 (Figure 2b) accompanied with a switch of water flow direction? If water flow is not consistent with the proposed modes of circulation or past circulation assumptions, it should be mentioned as well.

2. Line 127-131 - I think this would greatly benefit from a schematic of the proposed circulation modes.
3. Line 168-171 - This is messy. Figure 3 does not show data from FSW and FSE, and referring to the figure on line 170 made me more confused about what is on the Figure 3 (see also comment about figures). I think you are trying to say that in 2019 the density anomaly at FSs is higher than in FNs, thus Ronne HSSW came earlier to FSs, and it used to be the other way around. If it is what you are trying to say, then say so! Add a sentence at the end of this paragraph and tell the reader what the density difference means. Lastly, looking at Fig 3, I can not compare 27.90 and density values of FNE1 in 2019 (yellow end). Is 27.90 a better isopycnal to plot on Fig 3?
4. Lines 162-187. You mention "direct route", "less direct pathways" and "indirect pathway". Are the 2 or 3 pathways? Here is what I understood: 1) Direct= Site 5 -> FSW -> FNs 2) Less direct = Indirect = Site 5 -> FSE -> FNs. If this is correct, why does FSE show higher source salinity, and earlier signal (early 2017 vs late 2017) than FSW on Figure 2c? How does that work with "slower response" (line 187)? Perhaps this is aided by a schematic of proposed modes.
5. Do direct/indirect routes correspond to lines 281-285 in Janout et al? If so, how does their 2-6 years correspond to your estimates of 3-4 months to FSW ?

Minor comments

1. Line 95 - I found the word "maxima" confusing here. Maxima in depth? In time? Maybe "water above -2.2C is additionally present between..." ?
2. Lines 97-100 - Based on Figures 2a and 2b, I can see that temperature around -2.2C is present at approximately 550 m at the same time that there are peaks in the melt rate. First, is all of the water above -2.25C? It is not clear from the color bar. If it is not, I would like to see an isotherm -2.25C, and a comment about the possibility of freezing. Second, the temperature change at the interface in January - June 2017 is ~0.05C (this is what I can infer from the color bar), but the melt rate changes from 0.75 to 1.5 m/year. This seems a bit mismatched to me. How much does velocity change? Does warm water really "drive peak melt rates" or is it velocities? The radar is only 6.3 km away from FNE1, so I do not expect that ice shelf base is significantly lower there (making ice shelf base there be in contact with warm waters at ~600 m, for example).
3. Line 164 - "October/ November" - add year
4. Line 166 - Do these flow speeds include tides? Please comment (perhaps in the methods) how tides affect "total distance" that HSSW has to travel in order to get to Site 5, and how your time assessment may change.
5. I am not clear how you use the word "composite/composites". Is it one composite or many? Line 216 and lines 542-544 imply that each parameter gets its own composite. Figure 4a and line 222 imply that you use one composite.
6. Related to the definition of a composite, I am not clear what is plotted on Figure 4a. You have monthly mean data (line 543), and you plot it "for all months with enhanced SFRs" . You plot a mean of monthly means? Over what period? Is there a formula, similar to SFR?
7. Line 258 - I think reference to Figure 1c is incorrect.
8. I found "SFR" and "SFP" too close, and hard to keep track of. Unless they are both typical abbreviations used in literature, I suggest using SLP for pressure (as you do on line 276).
9. Line 330 - Something is wrong here.
10. Line 332 - 1985
11. Fig 1 (c,d) - What is the orange and green shading? I found no description for it in the figure caption. Is it for SFR values or for Ronne/Berkner modes (which are orange and green in Figures 1a, 2c, 3) ? If it is for SFR values, the shading does not match your analysis. Based on lines 191-193, I thought the shading should be for 2006-2015. Maybe another green shading for 1992-2003? If it is for Ronne/Berkner modes, point out where you talk about 2002(?) -2015 and 2015-2018.
12. Fig 2(b) - I can see 5 thick marks on the right vertical axis. Do they indicate the position of the instruments? If so, line 478 says that you have 6 instruments plotted here.

13. Fig 2(b) - it is not immediately obvious that this figure has time as a horizontal scale, and that matches subplots a, c and d. While I was able to infer it by referencing to dates in the text (line 92), this detracts from the paper.
14. Fig 3 - CTD profiles and mooring data are not labeled properly. Please make it easy to understand your figure.
15. Fig 3 (when compared to Figure 2b). Figure 2b shows that bottom water gets cooler (from yellow to blue). Fig 3 shows the same data, and it transforms from blue to yellow (time scale). Please change one of the color bars to add cohesiveness (presumably the time color bar).
16. Fig 4 (a) - 1) I think you have surface pressure as contours and wind speed as shading. 2) You introduce EBH and EWL, which are not defined (Yes, it is clear that one is a low, and one is a high, but please make it easy to understand your figure).

Reviewer #3 (Remarks to the Author):

Review: 'Recent atmospheric anomalies reduce threat of warm water to World's largest ice shelf' by Hattermann et al.

In this manuscript the authors report on measurements made via boreholes beneath the Filchner-Ronne ice shelf (FRIS), and link them to recent changes in the atmospheric circulation and sea ice production in the vicinity of the FRIS. They additionally link these anomalies to larger-scale climate indices, the Southern Annular Mode (SAM) and the depth of the Amundsen Sea Low (ASL), inferred from reanalysis data. The broader context for this investigation is that the FRIS is one of a few key sites around Antarctica that "fill" the global deep ocean with dense waters produced on the continental shelf. The FRIS has previously been shown to be vulnerable to warm water intrusions driven by climate change, which can irreversibly raise its melt rate by an order of magnitude and suppress deep water formation, but the potential for such a transition to occur in response to anthropogenic forcing is unknown.

The authors' measurements indicate a stronger throughflow of High Salinity Shelf Water (HSSW) from the face of the Ronne ice shelf through the FRIS cavity toward the front of the Filchner ice shelf. This intensified through flow coincides with anomalously rapid sea ice formation in front of the Ronne ice shelf, inferred from measured sea ice concentrations, bulk formulae and atmospheric reanalysis products. The anomalous sea ice formation, in turn, is driven by stronger winds blowing off the coast of the Antarctic continent in the western Weddell Sea. The authors show that such wind anomalies are correlated with variations in the SAM, lagging behind the SAM by ~10 months. They additionally infer that fluctuations in the longitude of the ASL can modulate the winds blowing offshore in the western Weddell Sea, potentially linking the dense water formation rates to equatorial and other climate variability via teleconnections.

This manuscript presents a remarkable set of measurements that provide a unique insight into the evolution of the circulation beneath the FRIS and its response to local and remote atmospheric drivers. The evidence for a change in the HSSW source at the front of the Filchner ice shelf is convincing, and does indeed strongly suggest a link to the authors' diagnosed changes in sea ice formation rates in front of the Ronne ice shelf. The authors acknowledge that there remains some ambiguity regarding this link, as they cannot account for internal variability in the cavity circulation.

However, I found the link between dense water formation rates and the SAM and ASL to be less convincing. While the SAM and the sea ice formation rate are correlated significantly, the SAM only explains around 25% of the variance in the in the sea ice formation when the long-term trend is removed. Thus although there is a relationship, clearly the sea ice formation is dominantly controlled

by other processes. The link to the ASL is very tenuous: as I understand it, the authors show that pressure anomalies in the Eastern Weddell are correlated significantly (albeit very weakly) with pressure anomalies in the Eastern Bellingshausen, and that pressure anomalies in the eastern Bellingshausen are correlated with the zonal position of the ASL (again, very weakly). From this they conclude that shifts in the ASL position produce anomalies in dense water formation in the Weddell. Clearly this is a very indirect link that I find very difficult to defend.

Overall, I think this paper should be published in Nature Communications, but it requires substantial revision. The authors need not change their analysis, but they do need to substantially revise their claims of links to the SAM and ASL to reflect the evidence presented. In addition, I have provided a list of additional minor and major comments/questions below.

Comments/questions:

The article's title is rather journalistic - more fitting of a press release on the authors' work than of a scientific article. Fair enough if the authors want to make the article accessible to a broad audience, but even in Nature the article titles tend to be written scientifically.

L41-42, L93, L97, L116, L234, L511: Mis-formatted symbols

L93-95: I had to read this sentence a few times to completely parse out the information. I suggest rephrasing for clarity: emphasize which drill sites are being referred to here, and where the HSSW that reaches them is sourced from.

L99: "Precise" should be more specific: either state how precise the measurement is, or omit this adjective.

Fig. 2: I was initially confused by the axes in this figure. I think it would help to repeat the month/year labels on the abscissas, particularly in panel B.

Also, the highlighted salinity bands in panel D seem to be much fresher than indicated in previous studies (e.g. Nicholls et al. 2009, Rev. Geophys.; Darelius and Saltee 2018, GRL).

L169-170 and Fig. 2 caption: I infer that this is surface-referenced potential density. Some care may be required in interpreting surface-referenced potential density changes/anomalies beneath the FRIS, as surface referenced potential density will substantially underestimate the contribution of temperature anomalies to density anomalies at these pressures.

L170-171: Please specify where this density difference reversal can be seen in the figures.

|

L166-181: Here the authors discuss the currents measured in the cavity and make inferences about circulation pathways. However, they only discuss the currents averaged over the full instrument deployment period in the shallowest and deepest instrument on each mooring, and only show the currents as arrows in Fig. 1. I am surprised by this because presumably there is a lot more to be gleaned from the current measurements than the authors are reporting here. In particular, do the current speeds evolve over time, similar to the salinities shown in Fig. 2C? A change in the currents would lend weight to the authors' inference of a strengthening of the cavity circulation following the enhanced sea ice formation starting in 2015.

L186-188: The salinities at FNE1, FNE2 and FSW also continue to increase as the salinity at Site 5 decreases. How can we reconcile these trends if the propagation time scale for Ronne HSSW through

the FRIS cavity is on the order of several months?

L193-201: As the authors note, the de-trended, 36-month averaged correlation coefficient is only ~ 0.5 , so only around 25% of the variance in SFR is explained by the SAM. This suggests that there is a connection, but rather a weak one. The authors should be more precise about the strength of this connection in here - saying that they "coincide" overstates that connection, in my opinion.

Fig. 4: The caption indicates that speed is indicated by contours, but I think the contours indicate pressure and the colors indicate speed.

L221-222: It is important to clarify here that SFR is defined using the time-mean wind speed. The full SFR, SFR*, estimated using the time-varying wind speed is relatively strongly correlated with both the open water area and with the wind speed itself.

L228: This is objectively not a strong correlation (10% of variance explained). In general I'm not sure that the evidence presented supports a strong link between the Weddell Sea and the ASL.

L234-235: Table S2 shows that the correlation between the ASL longitude and the EWS SLP is -0.03 , which seems to contradict this claim.

L254-255: Please clarify this statement. My reading was that Ronne-sourced ISW has not been observed in the Filchner trough previously, which is clearly not correct (e.g. Nicholls et al. 2009).

L266-268: Recent changes in the SAM are likely partially due to ozone forcing whereas future changes are predicted to be caused by greenhouse gas forcing (e.g. Shindell and Schmidt 2004, GRL). It is therefore unclear whether continuing increases in the SAM should support continued increases in the katabatic wind strength at the Ronne ice front, particularly if the air masses over the Antarctic continent begin to warm.

L272: Similar to my comments above, I really think this is overstating the link between the ASL and the dense water formation in the Weddell.

L469-471: In general the community seems to be shifting toward using the TEOS-10 standards i.e. conservative temperature and absolute salinity. I understand that using potential temperature and practical salinity allows more direct comparison with previous measurements, I would recommend that the authors also provide their results (perhaps as a supplement?) in terms of conservative temperature and absolute salinity, to facilitate direct comparison with future studies that use these thermodynamic variables.

L480-482: Do the currents also need to be corrected for tides?

L532-536: I'm not sure I fully understand the rationale here. It is reasonable to ask to what extent the SFR variations are associated with (wind-driven) changes in SIC, and setting U equal to its time mean is a reasonable way of separating these influences. However, I don't understand why SFR, rather than SFR*, is used as the primary quantifier of sea ice formation - isn't SFR* the more defensible estimate?

L538-547: There are some caveats associated with using reanalysis data around Antarctica that could be more strongly emphasized here. In particular, different products exhibit varying quality of agreement with measurements from meteorological stations (Bracegirdle and Marshall 2012, J. Climate), and varying decadal trends in wind speed and SLP (Hazel and Stewart 2019, J. Climate). In particular, the authors might consider switching to ERA5, rather than ERA-Interim (Dong et al. 2020, J. Climate).

L610: Spurious period after "<SFR>"

Fig. S2 caption: Do you mean "diamonds" rather than "stars"?

Author Response to the *reviewer's comments*

General response

We would like to thank all three reviewers for their careful and constructive assessment of our manuscript. Acknowledging the overall feedback, we have selected a new title “**Observed interannual changes beneath Filchner-Ronne Ice Shelf linked to large scale atmospheric circulation**” to better reflect the scope of our study. As a general response to the reviewer comments, in this revision we also shifted the focus from assessing the ice shelf’s response in future climate to the interannual changes that have been revealed by our observations and their links to the atmospheric forcing mechanisms. All reviewers provided important, constructive and often complementary input, and many clarifications that had been asked for could be addressed by the increased content allowance that the editor pointed out to us. In particular, we now include more explicit figure referencing, explain more carefully the inferred circulation pathways (including schematic arrows on the map and a new figure of the depth resolved ocean currents) and present a more clear and differentiated discussion of the links of the circulation change to large scale atmospheric forcing. Together with the point-by-point revisions and replies below, we are confident that our revised manuscript appropriately addresses the reviewer’s concerns and we kindly ask to consider its publication in Nature Communications.

Reviewer #1 (Remarks to the Author):

Main Comments

The manuscript by Hattermann et al is exciting and shows results that strongly improve our understanding of the Antarctic system. They show how remote atmospheric forcing (SAM and Amundsen Sea Low) regulates sea ice formation in the southern Weddell Sea, and consequently, basal melt of the largest ice shelf on the planet (Filchner Ronne Ice Shelf, FRIS). The new observations collected by the authors from beneath FRIS are unique and are the only ones that allow direct connections between remote forcing and ice shelf basal melt. The analysis is also accurate. The manuscript is relevant to the broader climate science community and therefore, I believe, appropriate for Nature Communications. I only have a couple of main comments that I believe need to be addressed for the manuscript to be consistent with what these new data show and with previous knowledge.

We thank the reviewer for this positive assessment. As will be explained in further detail below, we agree with the overall comments and have revised our manuscript to address them accordingly.

1) The overall message of the manuscript is a bit different compared to what the observations show. When I first read title and abstract I thought the manuscript would have shown “decadal to centennial results”, similar to Hellmer et al 2012. Instead, the new data, although incredibly important, are “only” few years long and making strong inferences and connections with processes that occur over much longer time scales (e.g. warming of the southern Weddell Sea) is probably too “audacious”. I would redirect the overall message (title and abstract) toward interannual variability and leave maybe one paragraph in the discussion to talk about long term implications. Interannual variability is key here and it has implication for basal melt and dense water production. Changes in these processes can have global consequences. Very important is the remote forcing on these interannual processes, as you describe.

We agree that we should more clearly distinguish between the physical processes identified in our data series and the long-term implications. In the revised version of the manuscript, we have removed the conjectures about a possible response of the FRIS system in a future climate. Instead, we present more carefully our observational dataset, the interannual variability seen in it, and the mechanistic relationship to atmospheric forcing. Our study stresses the influence of the atmospheric circulation on the circulation underneath the ice shelf and its implications for ice melt via its link to sea ice formation. The admittedly “short” data set is not sufficient to prove e.g. a general relationship between SAM/ASL and oceanic conditions, however, it shows that under the described atmospheric conditions the reaction of the circulation is the opposite of what has been expected so far (decreasing sea ice formation due to warming). How the atmospheric conditions will change in the future and the tropical influences on this, is not subject of our study, we only generally discuss possibilities. This should be reflected by changes throughout the entire manuscript.

2) Some important links to previous knowledge are missing in the present manuscript. First, Mckee et al 2012 (JGR, and other studies) show that changes in sea ice formation in the southern Weddell Sea can be associated with ENSO/SAM and this is not clearly expressed here. Second, there is a wide literature on the Antarctic Dipole (opposing sea level pressure in the Bellingshausen and Weddell seas) starting from Yuan and Martinson 2001 (GRL). Given the results shown here are associated with the Antarctic Dipole, a reference (and discussion) to these studies is important.

We agree that a discussion of the Antarctic Dipole and the teleconnections between the tropics and the polar ocean should be included here. We added information about the Antarctic Dipole and about influences of SAM/ENSO on sea ice (including the most important references) in the discussion section of the revised version of the manuscript. Also, in 2015/16 a strong El Nino event occurred. The location of the Bellingshausen Sea High in the periods of high sea ice formation rates found in our study would agree well with the general findings in various studies by Yuan and her co-workers. We added this in the text, too. However, as stated by Yuan et al. (2014) and other studies, the relationship between ENSO, SAM and sea ice is highly complex, thus we decided not to elaborate this in our study.

Third, increased sea ice formation has been argued to actually increase basal melt in cold ice shelf cavities (e.g. Nicholls 1997, Nature) because more "HSSW flushing" brings more heat into the cavity. This is the opposite of the message of this manuscript, and therefore a more detailed comparison with these previous results is required.

We agree with the reviewer that a differentiated discussion of this subject was missing in the previous version of the manuscript. Together with a more detailed presentation of the radar based melt rate estimates in the last paragraph of the first subsection in the results, we now include the statement that: "As melting deeper inside the FRIS cavity may be expected to increase under enhanced inflow of Ronne HSSW that brings heat deep into the cavity (Nicholls 1997), the mode shift that occurred in 2018 would indicate a shift in the balance of melting from the more northern Filchner Ice Shelf towards the grounding lines further south." We also general put less emphasis on the reduction in melting (by blocking of warm inflow) by the Ronne Mode, which makes this (at first sight apparent, but practically not really, see also comment below) contradiction less prominent.

Finally, HSSW formation is also influenced by glacial meltwater inflow (e.g. Silvano et al., 2018 Science Advances, Hellmer et al 2017, JCLIM etc.) and this process is not discussed here despite it seems relevant for the overall story.

We agree that this was not very carefully addressed in the previous version of the manuscript and we have augmented the discussion with the following paragraph:

“HSSW formation is also influenced by glacial meltwater inflow (Silvano et al. 2018). A sustained Berkner mode would eventually decrease the ISW density on the eastern part of the continental shelf, causing a potential feedback on the Berkner HSSW production and weakening the density barrier (Hellmer et al. 2017) that prevents mWDW from entering the ice shelf cavity (Darelius et al. 2016, Daae et al 2017). Although varying amounts of remotely sourced Ronne ISW and more locally produced Berkner HSSW have been observed in front of Filchner Ice Shelf (as indicated by blue and red letters in Fig. 1C), the density in the lower Filchner Trough has remained remarkably stable throughout recent decades (Janout et al, Nøst et al). This suggests that such feedbacks indeed exist, where the outflow of Ronne HSSW-sourced ISW acts as a precursor for the Berkner HSSW production (i.e. by determining the density at which convection that regulates HSSW production may occur). However, most of the meltwater from FRIS is currently exported through Filchner Trough and does not affect the western continental shelf regions where the densest HSSW is produced that drives the Ronne mode. This indicates that the polynya activity in front of Ronne Ice Shelf really is the main regulator of the circulation of the southern Weddell Sea continental shelf.”

I would introduce/discuss this previous knowledge in the introduction/discussion and put strong emphasis on the amazing new data shown here (from the ice shelf cavity). Only these new data provide definitive evidence of the remote connection between the atmosphere and the “dark” ice shelf cavities.

We are encouraged by this supportive assessment and have revised the abstract, the last paragraph of the introduction, as well as much of the results and discussion sections to better present the new insights from our dataset and their relation to previous knowledge.

Minor comments

- *Figure 1: Specify in the caption what the “orange and green periods” are in panel C and D. Also mention what “upper” and “lower” ocean currents mean.*

The following explanations have been added to the figure caption:

“Vectors indicate time averaged current meter observations from the lowest (black) and highest (white) instrument in the water column and legends showing a 2 cm s⁻¹ (10 cm s⁻¹) scale for the Filchner sites (Site 5). Color shaded areas in C & D indicate the 15 year period (orange and green) relative to which the spatial pattern of SFR anomalies (green only) is computed that is shown in A.”

- *Line 145-147: I would refer to Fig. 2A here to highlight the high melt rate in 2016.*

The reference has been added. In addition, we consolidated all melt rate discussion in the last paragraph of the first subsection of the results for clarity.

• *Line 147-150: this sentence is a bit unclear, or at least not explained. Please provide some further explanation of why more HSSW can cause increased basal melt deeper in the cavity. This also opposes the main message of the manuscript (more HSSW less melting). Moreover, in terms of ice sheet stability, the deeper portions of the cavity are actually more relevant. See also the main comments.*

While an explanatory sentence has been added that HSSW brings heat into the cavity, the change between the Ronne mode and the Berkner mode may affect melt rates beneath FRIS in two different ways. Generally a “healthy” Ronne mode is needed to stop the flip into a different regime in which warm water comes south to the Filchner Ice Shelf and causes very high melt rates. But more Ronne Mode also means more deep melting near grounding lines, which might reduce GL stability. But at least a lot of that melt will end up back on the ice shelf base, even if not in a place that helps ice shelf stability too much. Increasing Berkner Mode increases melt further north, nearer the ice front. How much that affects buttressing is unclear, but there have been modelling studies, as e.g. Fürst et al. (2017) highlighting the importance of the northern parts of the ice shelves. The other point is that increasing Berkner Mode only directly affects melting further north on Filchner. Not Ronne. And then, there is the possibility that both act together: increasing BM and increasing RM. All that says is that a reduction in RM (and more generally with RM or BM) leads to a reduction in melting that would have been associated with those HSSW inflows, at the respective locations these occur. During our analysis, we tried to more carefully assess these changes in melt pattern, e.g. by using remote sensing datasets, such as was recently presented by Adusumili et al (2020). However, since no robust relationships could be established, beyond the result shown from the melt rate radar in the manuscript, we prefer to limit the discussion of these potential responses.

• *Line 152-188: It might be a good idea to include in a map the circulation pathways established in this study.*

A similar comment was also made by reviewer #2 and we now include “Colored arrows [that] indicate inferred pathways and transformation from HSSW into ISW in the Ronne mode (orange to blue) and Berkner mode (green to purple).” In Fig. 1A. We did consider including another figure with schematic cross sections as the ones shown below, but found benefits to be marginal compared to the space it would take. However, if the reviewers find it still necessary to include such visualization of the circulation modes (which really play out in 3 dimensions), we may add it at a later stage.

Fig. R1: Schematic cross section of the circulation beneath FRIS in the Berkner mode (left) and Ronne mode (right). Note that the third dimension is collapsed, such that Ronne HSSW (green arrow) would usually from the Ronne Ice Front, while Berkner HSSW (red arrow) enters the cavity at the Filchner Ice Front.

- Line 201: *Maybe this is Fig S2.*

Yes, we corrected that.

- Line 211: *“Large scale climate indicators”.*

We adopted that terminology in our revision of that part of the manuscript, now being mentioned in the beginning of the third paragraph of the last subsection of the results.

- Line 242: *sea ice moves roughly aligned with the winds (e.g. Holland and Kwok, 2012, Nat. Geosci.). So the Ekman forcing might not be relevant here.*

We fully agree that the sea ice generally follows the atmospheric flow, as Holland and Kwok (2012) find in their study, which is, in Antarctica, clockwise around the center of the cyclones, i.e. northward movement at the western side and southward movement at the eastern side of the cyclone; and sea ice extent anomalies can be mostly explained by the prevailing atmospheric flow. Since this is usually a circular movement, a small angle between the ice movement and the wind direction is of minor importance here. However, in a strong zonal westerly flow, which is common at the northern edge of the circumpolar trough, ice is displaced northward due to the angle between wind direction and direction of the ice flow, which is particularly important when SAM is in a strongly positive phase with strong zonal winds. Various studies, (including Holland and Kwok (2012)), state that there is a “turning angle” between wind and movement of the ice (e.g. Holland and Kwok (2012), Nansen (1897, 1902), Kottmeyer et al. (1997), Maeda et al. (2020), Kimura (2004)).

In the revised version of the manuscript, we included a more differentiated discussion of this mechanism, now also highlighting a few noteworthy anomalies in the Atlantic sector of the Southern Ocean during the period when enhanced SFRs reinforced the Ronne mode under FRIS.

- *Line 261: Rapid changes have been observed in the western peninsula, while on the eastern peninsula (closer to FRIS) changes are less "strong" (except for ice shelf disintegration).*

We have removed that statement when revising the focus of the discussion as described in the general reply to the reviewers comments.

Reviewer #2 (Remarks to the Author):

General Comments

The paper presents new observations from the Weddell Sea and the Filchner-Ronne Ice Shelf cavity and an analysis of the observations. The results are very interesting not only for the community focused on FRIS, but also for the wider ice shelf-ocean community. I found the analysis of diverse observations clever and fascinating. I think the paper will be a great addition to the knowledge about ice shelf cavities after appropriate corrections are made.

We thank the reviewer for this positive assessment and the constructive comments that helped to improve our manuscript.

The paper describes the following process. Recent changes in the sea ice production, caused by atmospheric anomalies, lead to a new circulation pattern in the FRIS cavity. As a result of this circulation,

Filchner Ice Shelf frontal region experiences outflow of Ice Shelf Water, traced back to the Ronne polynya, and not ISW produced locally. This outflow changes the structure of the water column in the Filchner Trough.

As mentioned in our general response, and in specific replies to reviewer #1, we fully agree with this summary of the scope of our work.

I found the title leaning towards “clickbait”. Although I can see it being very attractive, I feel that the paper did not match the expectations set by the title and I suggest changing the title. Two main concerns arise here. First, your analysis is nuanced and well-constrained, and suggests that no long-term effects on warm water inflow can be predicted. Thus “reduce threat” feels like an oversell to me. Second, I feel like the paper did not talk about warm water enough to be reflected in the title. You describe a new circulation pattern in the cavity, and contrast it to one modelling study (reference 13). I think it is very important to talk about implications of the new circulation pattern, as you do in the last section on the paper, but this is not the main result of your paper.

We agree with the reviewer and have selected a new title to better reflect the scope of our study.

To justify the title, I think one has to present some evidence of warm waters inflow in the Filchner Trough, and the reversal of the signal in recent years.

Originally, we were confident that the principle of the density barrier was sufficiently established through recent modelling studies (Hellmer et al. 2017, Hazel et al. 2020, Daae et al. 2020). Observations of modified Warm Deep Water near the Filchner Ice Shelf front (Darelius et al. 2016) confirmed that these warm waters are too light to enter the cavity, while a sustained Berkner Mode, without salt input from the Ronne polynya, would eventually cause freshening in the Filchner Trough. However, we realize that these conjectures were mainly derived from theoretical considerations. Hence, as explained in the general response, we have revised the abstract, the last paragraph of the introduction, as well as much of the results and discussion sections to better present the new insights from our dataset and their relation to previous knowledge.

Additionally, while you mention that FRIS is the largest ice shelf by volume (line 39), the title “largest ice shelf” is typically used for the Ross Ice Shelf.

We removed that statement to avoid ambiguity.

In general, the paper reads as rushed, and many small inconsistencies make it hard to appreciate the science. A lot of work has to be put into reading the paper and deciphering the relationship between text and figures. In my opinion, the authors did not make the material easy to understand.

We acknowledge this assessment and worked on improve the presentation of the science throughout our revision. In part, this was possible by including more explicit figure references and more detailed explanations (including a new figure presenting the current meter observations beneath the ice shelf) within the increased length allowance of the journal. We also followed the reviewer's remarks to identify where improvements and clarity were needed. Eventually, some rearrangements were made to better group different aspect of the observations (e.g. moving all presentation of the melt rate radar results at the end of the first subsection in the results section). Hopefully, these measures have helped to improve the presentation of our work.

Major Comments

1. I am confused about your choice to talk about 2 modes of circulation, and not discuss observed velocities more (first at lines 115-122, and elsewhere). Did water flow direction change at FSs during 2017? Did water flow direction change at Site 5 (possibly twice, in 2017 and 2018)? Having multiple velocity measurements throughout the water column, could you see the difference in how top and bottom layers reacted? Is cooling in the bottom of FNE1 (Figure 2b) accompanied with a switch of water flow direction? If water flow is not consistent with the proposed modes of circulation or past circulation assumptions, it should be mentioned as well.

As now being stated explicitly in first subsection of the results in the revised version of the manuscript: "based on [the] shift in source water mass properties, two circulation modes may be defined to be dominant under northern Filchner Ice Shelf. [...] While being most visible from changes in temperature and salinity that contain information about the history of the water masses, also the current meter records confirm a shift from a largely partitioned cavity circulation where local inflows dominate the northern Filchner Ice Shelf (Berkner Mode), toward a coherent circulation that spans the entire FRIS in response to increased HSSW inflow at the Ronne ice front (Ronne Mode)," as is now being explicitly discussed together with a new figure in the second subsection of the results.

Generally, we find it challenging, based a handful of locations with velocity measurements, to constrain the cavity-wide (mainly thermohaline driven) circulation under an ice shelf that covers an area comparable with the size of Sweden. In particular under the Filchner Ice Shelf, "current meter measurements [...] are generally dominated by strong tidal currents that can exceed 50 cm/s during spring tides. This indicates a rather diffusive flow regime, in which precise transport estimates are challenging and the thermohaline mean flow velocities only contribute as small residuals" (second subsection of the results). Ocean currents are susceptible to local topography and other processes that

affect the flow at smaller scales. Using properties of the water masses to trace back their origin, quickly becomes a more powerful tool in this configuration, in particular where the ice shelf provides a semi-enclosed system, with heat and salt fluxes are largely constrained to the ice-ocean interactions.

We agree with the reviewer that our presentation may have appeared confusing due to lack of explanation. However, the choice of defining the two circulation modes based on the observed pronounced and persistent change of water masses in the Filchner Trough (together with the time averaged mean flow directions), appears most robust to us, while the depth resolving current meter time series give a more comprehensive picture that lends additional support. In fact, depending on the scale of focus, there are several more aspects that are revealed by the current meter and hydrographic time series from under FRIS, but that would by far exceed the scope of this study. Hopefully the reviewer will agree that we found a useful balance for the level of detail that is being presented in the revised version of the manuscript.

2. Line 127-131 - I think this would greatly benefit from a schematic of the proposed circulation modes.

In addition to the more explicit definition of the two circulation modes: “A “Berkner mode” where the Filchner Trough is filled with Berkner HSSW that enters the cavity at the Filchner ice front and determines the ISW properties under northern Filchner Ice Shelf, and a “Ronne mode”, where remotely sourced ISW, originating from HSSW inflows at the Ronne ice front, propagates through the FRIS cavity toward the northern Filchner Ice Shelf to dominate water mass properties in the Filchner Trough.”, we include the pathways of these water masses on the map in Fig 1. of the revised version of the manuscript. As also stated in our response to reviewer #1, we may also add schematic of the vertical cross section to illustrate how the Ronne mode and Berkner mode interfere at the Filchner Ice Shelf front, if the reviewer thinks this would still be necessary after the changes that are already implemented.

3. Line 168-171 - This is messy. Figure 3 does not show data from FSW and FSE, and referring to the figure on line 170 made me more confused about what is on the Figure 3 (see also comment about figures). I think you are trying to say that in 2019 the density anomaly at FSs is higher than in FNs, thus Ronne HSSW came earlier to FSs, and it used to be the other way around. If it is what you are trying to say, then say so! Add a sentence at the end of this paragraph and tell the reader what the density difference means. Lastly, looking at Fig 3, I can not compare 27.90 and density values of FNE1 in 2019 (yellow end). Is 27.90 a better isopycnal to plot on Fig 3?

A time series of the density evolution at the southern and northern Filchner sites is included as part of the additional figure in the revised version of the manuscript, showing that: “Consistent with an intensified eastward propagation of saline Ronne HSSW through the FRIS cavity, potential density anomalies at FSW and FSE increase from below 32.68 kg m⁻³ in 2016 to above 32.69 kg m⁻³ in 2018, eventually causing a reversal of the, the density difference between the southern and northern Filchner

drill sites (Fig. 4E).”, here also adopting TEOS-10 potential density referenced to the approximate deployment pressure at the instruments of 1000 dbar.

4. Lines 162-187. You mention “direct route”, “less direct pathways” and “indirect pathway”. Are the 2 or 3 pathways? Here is what I understood: 1) Direct= Site 5 -> FSW -> FNs 2) Less direct = Indirect = Site 5 -> FSE -> FNs. If this is correct, why does FSE show higher source salinity, and earlier signal (early 2017 vs late 2017) than FSW on Figure 2c? How does that work with “slower response” (line 187)? Perhaps this is aided by a schematic of proposed modes.

5. Do direct/indirect routes correspond to lines 281-285 in Janout et al? If so, how does their 2-6 years correspond to your estimates of 3-4 months to FSW?

We clarified the description of the direct and less direct pathway as follows in the revised version of the manuscript: “Alongside the direct pathway revealed by the seasonal propagation of HSSW pulses from northern Ronne Ice Shelf via Site 5 along the western flank of the Filchner Ice Shelf cavity to FSW (Fig 1A), the gradual increase in source water salinities at FSE on the eastern flank (light red curve in Fig. 2C) witnesses a less direct and slower pathway throughout the cavity. Source water salinities at FSE are generally higher than at FSW and start to increase before the distinct pulse arrives at Site 5 in mid-2017. This suggests an influence of Ronne HSSW at FSE that has been cycled deeper inside the cavity, probably originating from a previous season, as is supported by tracer analyses that suggests an interannual propagation time scale throughout the FRIS cavity (Janout et al. in rev.). Such a delayed response to HSSW inflows along less direct pathways also explains that source water salinities at all moorings under Filchner Ice Shelf continue to rise, when source water salinities at Site 5 gradually decline during 2018/19, likely due to fewer periods of high sea ice formation during 2018 compared with the previous two years (Fig. 2D). Ronne sourced ISW that eventually arrives at the Filchner Ice Shelf front hence represents the integrated signal from those different pathways.”

Minor comments

1. Line 95 - I found the word “maxima” confusing here. Maxima in depth? In time? Maybe “water above -2.2C is additionally present between...” ?

The formulation has been revised as follows: “While the warmest waters appear close to the seafloor, pulses of elevated temperatures between 600 and 800 m indicate inflows into the cavity also at intermediate density/depth levels”

2. Lines 97-100 - Based on Figures 2a and 2b, I can see that temperature around -2.2C is present at approximately 550 m at the same time that there are peaks in the melt rate. First, is all of the water

above -2.25C? It is not clear from the color bar. If it is not, I would like to see an isotherm -2.25C, and a comment about the possibility of freezing. Second, the temperature change at the interface in January - June 2017 is ~0.05C (this is what I can infer from the color bar), but the melt rate changes from 0.75 to 1.5 m/year. This seems a bit mismatched to me. How much does velocity change? Does warm water really “drive peak melt rates” or is it velocities? The radar is only 6.3 km away from FNE1, so I do not expect that ice shelf base is significantly lower there (making ice shelf base there be in contact with warm waters at ~600 m, for example).

We included the -2.25°C isotherm in the figure as was requested by the reviewer, together with the following description of the time series in the revised version of the manuscript: “The ocean cooling beneath northern Filchner Ice Shelf during the transition into the Ronne Mode is accompanied by a decrease in local basal melting as observed by downward-looking radars that were deployed near FNE1 in January 2016, and then at FNE1 from the time of the mooring installation (Methods, Fig. 2A). Considering the environment of relatively strong tidal mixing beneath FRIS (Makinson et al.), and comparing with the pressure-dependent melting point temperature of ≈ -2.25 °C beneath the 500-600 m deep ice base in this region, these relatively warm water intrusions provide heat to increase local melting at the ice base. Consistently, peak melt rates of over 1.5 meters per year, coincide with peak temperatures observed at the two uppermost instruments at FNE1 in 2017, while melt rates drop to lower levels when temperatures permanently stay below -2.2°C at all sensors in 2018.” The magnitude of the currents is determined by the strong tides in that region and does not change significantly on the longer time scales over which the melt rate variability is observed. Since the uppermost instrument is deployed only few meters below the ice base, we do not expect temperatures to increase much above the in situ melting point of the nearby ice base, as heat will quickly be consumed by melting in this tidally mixed environment. Proper melt rate estimates would need to account for the vertical fluxes that are present at a given equilibrium temperature in the boundary layer under the ice. For the same reason, the increased temperatures seen in the entire water column in 2017 will contribute to increase the melt rates, because the heat will be made accessible to the ice ocean interface through tidal mixing. We have not found any indications of freezing at that site. Indications for freezing have not been observed at that site.

3. Line 164 - “October/ November” - add year

Together with an additional figure of the temperature time series, we specified in the revised version of the manuscript that: “Seasonal temperature and maxima occurring each year in October/November at FSW (Fig. 4C) suggest that Ronne HSSW pulses originating from mid-winter peak sea ice formation arrive with a three to four month lag east of Berkner Island.”

4. Line 166 - Do these flow speeds include tides? Please comment (perhaps in the methods) how tides affect “total distance” that HSSW has to travel in order to get to Site 5, and how your time assessment may change.

As mentioned in the revised version of the manuscript, these estimates are based on the low-pass filtered residual mean currents. Tides will add multiple components, such as a diffusive transport at length- and time scales over tidal excursion cycles, as well as effects as Lagrangian tidal rectification. Those contributions are not primarily measurable with our instruments and a quantitative assessment is challenging. It may also be that some of the residual flow is a result of Eulerian tidal rectification in the frictional boundary layers over sloping topography. We have included the following more detailed explanation together the assessment of transportation time scales in the revised version of the manuscript, in combination with the figure of current meter time series:

“The arrival of those Ronne HSSW pulses is also seen as a seasonal increase in low-pass filtered southward flow speeds that vary between 10-15 cm s⁻¹ at the upper and lower current meter deployed at Site 5 (Fig. 4A). Current meter measurements beneath Filchner Ice Shelf are generally dominated by strong tidal currents that can exceed 50 cm/s during spring tides. This indicates a rather diffusive flow regime, in which precise transport estimates are challenging and the thermohline mean flow velocities only contribute as small residuals. However, while velocities at the upper current meters at the southern Filchner sites (Fig. 4C & D) are more dominated by local processes, the low pass filtered data from the lower instruments show residual currents in the order of a few cm/s, confirming the northward propagation of Ronne HSSW-sourced ISW along both flanks of the southern Filchner Ice Shelf (Fig 1A). Seasonal temperature and maxima occurring each year in October/November at FSW (Fig. 4C) suggest that Ronne HSSW pulses originating from mid-winter peak sea ice formation arrive with a three to four month lag east of Berkner Island. Considering the observed flow speeds, this is a plausible time scale for the HSSW signal to be advected along an approximately 800 km long pathway from Ronne Ice Front to FSW, although with some uncertain about contributions from the tides”

5. I am not clear how you use the word “composite/composites”. Is it one composite or many? Line 216 and lines 542-544 imply that each parameter gets its own composite. Figure 4a and line 222 imply that you use one composite.

We added the following explanation in the revised version of the manuscript: “To assess which atmospheric conditions may promote the Ronne mode beneath FRIS, monthly anomalies of surface pressure (SP), wind speed and direction derived from ECMWF ERA Interim²⁴ were averaged over all months with enhanced SFR events indicated in Fig 1D. The resulting composite (Fig. 5A) shows that...”

6. Related to the definition of a composite, I am not clear what is plotted on Figure 4a. You have monthly mean data (line 543), and you plot it “for all months with enhanced SFRs”. You plot a mean of monthly means? Over what period? Is there a formula, similar to SFR?

As mentioned in the reply to the previous comment, the composite is obtained by averaging the monthly anomalies of the atmospheric variables for all months where enhanced SFR events occurred.

7. Line 258 - I think reference to Figure 1c is incorrect.

We have specified in the revised version of the manuscript that: “varying amounts of remotely sourced Ronne ISW and more locally produced Berkner HSSW have been observed in front of Filchner Ice Shelf (Janout et al. under review) (as indicated by blue and red letters in Fig. 1C)”, and as is also explained in the figure legend.

8. I found “SFR” and “SFP” too close, and hard to keep track of. Unless they are both typical abbreviations used in literature, I suggest using SLP for pressure (as you do on line 276).

Sea Level Pressure (SLP) would be undefined over the Antarctic continent. Our analysis is based on Surface Pressure and we have changed the abbreviation to “SP” accordingly.

9. Line 330 - Something is wrong here.

The corrupt reference has been corrected in the revised version of the manuscript

10. Line 332 – 1985

The year in the reference has been corrected.

11. Fig 1 (c,d) - What is the orange and green shading? I found no description for it in the figure caption. Is it for SFR values or for Ronne/Berkner modes (which are orange and green in Figures 1a, 2c, 3) ? If it is for SFR values, the shading does not match your analysis. Based on lines 191-193, I thought the shading should be for 2006-2015. Maybe another green shading for 1992-2003? If it is for Ronne/Berkner modes, point out where you talk about 2002(?) -2015 and 2015-2018.

We have added: “Color shaded areas in C & D indicate the 15 year period (orange and green) relative to which the spatial pattern of SFR anomalies (green only) is computed that is shown in A.” to the figure caption, as well as the following statement in the revised version of the manuscript: “A map of SFR anomalies (Fig 1A) shows that heat loss in the Ronne polynya from 2015 to 2018 on average was more than 20 Wm⁻² higher than the average of the last 15 years (2003 to 2018, blue and orange patch in Fig. 1C & D).”

12. Fig 2(b) - I can see 5 thick marks on the right vertical axis. Do they indicate the position of the instruments? If so, line 478 says that you have 6 instruments plotted here.

There are five instruments deployed at FNE1. We corrected that mistake in the methods section in the revised version of the manuscript.

13. Fig 2(b) - it is not immediately obvious that this figure has time as a horizontal scale, and that matches subplots a, c and d. While I was able to infer it by referencing to dates in the text (line 92), this detracts from the paper.

We added the time axis in the panel in the figure in the revised version of the manuscript.

14. Fig 3 - CTD profiles and mooring data are not labeled properly. Please make it easy to understand your figure.

We have added complete labels for the CTD profiles and mooring data in the figure in the revised version of the manuscript.

15. Fig 3 (when compared to Figure 2b). Figure 2b shows that bottom water gets cooler (from yellow to blue). Fig 3 shows the same data, and it transforms from blue to yellow (time scale). Please change one of the color bars to add cohesiveness (presumably the time color bar).

We have inverted the color bar on the time axis in Fig. 3 of the revised version of the manuscript according to the reviewer's recommendation.

16. Fig 4 (a) - 1) I think you have surface pressure as contours and wind speed as shading. 2) You introduce EBH and EWL, which are not defined (Yes, it is clear that one is a low, and one is a high, but please make it easy to understand your figure).

We have corrected the labels on the figure, which should be EWS and EBS as consistently introduced in the text.

Reviewer #3 (Remarks to the Author):

Review: 'Recent atmospheric anomalies reduce threat of warm water to World's largest ice shelf' by Hattermann et al.

In this manuscript the authors report on measurements made via boreholes beneath the Filchner-Ronne ice shelf (FRIS), and link them to recent changes in the atmospheric circulation and sea ice production in the vicinity of the FRIS. They additionally link these anomalies to larger-scale climate indices, the Southern Annular Mode (SAM) and the depth of the Amundsen Sea Low (ASL), inferred from reanalysis data. The broader context for this investigation is that the FRIS is one of a few key sites around Antarctica that "fill" the global deep ocean with dense waters produced on the continental shelf. The FRIS has previously been shown to be vulnerable to warm water intrusions driven by climate change, which can irreversibly raise its melt rate by an order of magnitude and suppress deep water formation, but the potential for such a transition to occur in response to anthropogenic forcing is unknown.

The authors' measurements indicate a stronger throughflow of High Salinity Shelf Water (HSSW) from the face of the Ronne ice shelf through the FRIS cavity toward the front of the Filchner ice shelf. This intensified through flow coincides with anomalously rapid sea ice formation in front of the Ronne ice shelf, inferred from measured sea ice concentrations, bulk formulae and atmospheric reanalysis products. The anomalous sea ice formation, in turn, is driven by stronger winds blowing off the coast of the Antarctic continent in the western Weddell Sea. The authors show that such wind anomalies are correlated with variations in the SAM, lagging behind the SAM by ~10 months. They additionally infer that fluctuations in the longitude of the ASL can modulate the winds blowing offshore in the western Weddell Sea, potentially linking the dense water formation rates to equatorial and other climate variability via teleconnections.

We adhere to this summary of the scope and content of our study.

This manuscript presents a remarkable set of measurements that provide a unique insight into the evolution of the circulation beneath the FRIS and its response to local and remote atmospheric drivers. The evidence for a change in the HSSW source at the front of the Filchner ice shelf is convincing, and does indeed strongly suggest a link to the authors' diagnosed changes in sea ice formation rates in front of the Ronne ice shelf. The authors acknowledge that there remains some ambiguity regarding this link, as they cannot account for internal variability in the cavity circulation.

We appreciate this positive assessment and the value seen in our work. We also thank the reviewer for the constructive comments below, regarding the robustness of the relation between the sea ice formation rates and the large scale atmospheric circulation. We acknowledge that some links or claims

may have been presented overratedly, although we also realized that some causalities that we deduce from our analysis had not been presented clearly enough, as clarified in greater detail below.

However, I found the link between dense water formation rates and the SAM and ASL to be less convincing. While the SAM and the sea ice formation rate are correlated significantly, the SAM only explains around 25% of the variance in the in the sea ice formation when the long-term trend is removed. Thus although there is a relationship, clearly the sea ice formation is dominantly controlled by other processes.

We fully agree that the influence of SAM should be handled with caution and not overplayed. SAM explains only about one third of the atmospheric variability and even less of the variance in sea ice. Particularly, SAM gives no information about the longitudinal position of the pressure anomalies that are highlighted by our composite analysis. A longer data set would be necessary for a more thorough investigation. However, this is not the focus of our study. Following also the advice of Reviewer #1, we tried to make it clearer that the focus of our study is the response of the circulation underneath the ice shelf to the atmospheric forcing. Our study shows that under the described atmospheric conditions the response of the circulation beneath the ice shelf is governed by a reinforcement of the density driven circulation, which is the opposite of what has been expected for a warming climate in more recent modelling studies (Hellmer et al, 2017).

The link to the ASL is very tenuous: as I understand it, the authors show that pressure anomalies in the Eastern Weddell are correlated significantly (albeit very weakly) with pressure anomalies in the Eastern Bellingshausen, and that pressure anomalies in the eastern Bellingshausen are correlated with the zonal position of the ASL (again, very weakly). From this they conclude that shifts in the ASL position produce anomalies in dense water formation in the Weddell. Clearly this is a very indirect link that I find very difficult to defend.

It is true that the correlation between the pressure anomalies in the Eastern Weddell Sea and the ASL is only weakly significant. We believe, however, that maybe, in our submitted manuscript, we did not explain the mechanism we want to present clearly enough: We did not intend to suggest a causal relation between pressure anomalies in the Eastern Weddell Sea and the position of the ASL. However, the position of the ASL is necessarily related to the existence (or non-existence) of an anticyclone in the Eastern Bellingshausen Sea ($r(\text{ASL}|\text{lon EBS}) = -0.52$), Table S1) and thus the direction of the general flow across the ice-shelf edge (Fig 4 A), which directly influences sea ice formation and thus dense water production. The negative pressure anomalies in the Eastern Weddell Sea are an additional factor (loosely related to SAM index as discussed above) that strengthens the offshore winds over the Ronne Ice Shelf. During the strong sea ice formation event in 2016, both factors were present (yellow bars in Fig 4 B & C) together with an westward displaced ASL and a strong positive SAM index (blue curves in Fig 4 B & C). Although the statistical evidence for this connection on the short time series is weak, we believe that that the mechanism is plausible and deserves attention, because it gives the atmospheric

projections predictive power , i.e., if we know how the remote atmospheric patterns are likely to change (through coupled climate models), we can make predictions about future conditions in the FRIS cavity. As mentioned above, we rephrased this in the text to clarify it. Longer-term implications can only be discussed qualitatively here and more exact simulations of the atmospheric conditions are necessary to yield statements about the future of the ice shelf. The above discussion has been added to the third subsection of the results in the revised version of the manuscript.

Overall, I think this paper should be published in Nature Communications, but it requires substantial revision. The authors need not change their analysis, but they do need to substantially revise their claims of links to the SAM and ASL to reflect the evidence presented. In addition, I have provided a list of additional minor and major comments/questions below.

We substantially revised the third subsection of the results and the discussion sections to address the reviewer's concerns. In particular, we utilized the extended length allowance of the journal (the original submission was forwarded from a shorter formatted journal) to provide a more differentiated discussion of the robustness and plausibility of the links between the sea ice formation rates in front of the Ronne Ice Shelf and the large scale atmospheric circulation. In addition, we hope that the revised version of the manuscript also more clearly describes the mechanisms that we would like to put forward.

Comments/questions:

The article's title is rather journalistic - more fitting of a press release on the authors' work than of a scientific article. Fair enough if the authors want to make the article accessible to a broad audience, but even in Nature the article titles tend to be written scientifically.

We agree with the reviewer and have selected a new title to better reflect the scope of our study.

L41-42, L93, L97, L116, L234, L511: Mis-formatted symbols

These came in during the upload process on the journals submission site. We apologize for not noticing prior to submission and hope to be able to avoid mistakes upon resubmission.

L93-95: I had to read this sentence a few times to completely parse out the information. I suggest rephrasing for clarity: emphasize which drill sites are being referred to here, and where the HSSW that reaches them is sourced from.

We have included the following statement in the first paragraph of the result section to clarify the drill site location and origin of the inflows in the revised version of the manuscript: “The first 9-12 months of the ocean time series (starting in December 2016) from the northern drill sites on Filchner Ice Shelf (FNE1 & FNE2, Fig. 1A) show temperatures close to the surface freezing point (-1.9 °C, Fig. 2B), indicating the direct inflow of HSSW at least as far as 40 km inland of the calving front.”

L99: “Precise” should be more specific: either state how precise the measurement is, or omit this adjective.

We followed the reviewer’s advice and omitted this adjective in the revised version of the manuscript.

Fig. 2: I was initially confused by the axes in this figure. I think it would help to repeat the month/year labels on the abscissas, particularly in panel B.

This was also pointed out by reviewer #2, we have added a time axis on that panel in the revised version of the manuscript.

Also, the highlighted salinity bands in panel D seem to be much fresher than indicated in previous studies (e.g. Nicholls et al. 2009, Rev. Geophys.; Darelius and Sallee 2018, GRL).

The source water salinity bands that were highlighted in Fig. 2D of the original submission were derived from the water mass end-member types shown in Fig. 3 of the same manuscript. There, the Gade line that connects the Ronne HSSW-derived ISW end member with the surface melting point is centered on around 34.76 practical salinity units while the Berkner HSSW end member is centered on around 34.69 practical salinity units. A comparison with Fig 11 in Nicholls et al. 2009 shows that densest water masses in the Ronne depression (60 W) have salinities between 34.70 and 34.80, whereas salinities on Berkner Bank (50 W) can be as low as 34.60 and do not exceed 34.75, which seems to be consistent with our results. Fig. 11 in Nicholls et al. 2009 shows that extreme values of above 34.8 practical salinity units have been measured, but those extreme source water salinities have never been observed in the Filchner Trough (a detailed analysis of historical CTD data in the Filchner Trough was part of our analysis, and may be provided to the reviewer upon request), most likely because of dilutive mixing along the pathway through the cavity. Note also that any subglacial runoff from the grounded ice sheet into the ice shelf cavity would shift the Gade line towards lower salinities. The source water salinity bands in Darelius & Sallee (2018) are based on conservative temperature and absolute salinity, which were not comparable in the original submission, while after converting to TEOS-10 our values of between 34.85

g/kg for the Berkner HSSW source and around 34.92 g/kg for the Ronne HSSW source match rather well with the estimates of Darelius & Sallee who quote the same numbers in their figure 4.

L169-170 and Fig. 2 caption: I infer that this is surface-referenced potential density. Some care may be required in interpreting surface-referenced potential density changes/anomalies beneath the FRIS, as surface referenced potential density will substantially underestimate the contribution of temperature anomalies to density anomalies at these pressures.

This is a valid concern, and while we had previously assessed the robustness of our results with respect to pressure effects on the density, we agree with the reviewer that this should be clarified in the manuscript. The new figure 4 in the revised version of the manuscript now includes time series of potential density at the lower instruments at all Filchner sites referenced to 1000dbar to avoid any ambiguity.

L170-171: Please specify where this density difference reversal can be seen in the figures.

In panel E of the new figure 4 in the revised version of the manuscript, as is also explained in the text.

L166-181: Here the authors discuss the currents measured in the cavity and make inferences about circulation pathways. However, they only discuss the currents averaged over the full instrument deployment period in the shallowest and deepest instrument on each mooring, and only show the currents as arrows in Fig. 1. I am surprised by this because presumably there is a lot more to be gleaned from the current measurements than the authors are reporting here. In particular, do the current speeds evolve over time, similar to the salinities shown in Fig. 2C? A change in the currents would lend weight to the authors' inference of a strengthening of the cavity circulation following the enhanced sea ice formation starting in 2015.

See also our response to reviewer #2 regarding the discussion of velocity time series. There is certainly much to be learnt from the velocities, but those insights are partially peripheral and may distract from the story in this paper. Hence, the currents were initially left out for simplicity, but are now included in the revised version of the manuscript, together with a more careful discussion in the second subsection of the results as to what extent they support our inferences based on water mass properties.

L186-188: The salinities at FNE1, FNE2 and FSW also continue to increase as the salinity at Site 5 decreases. How can we reconcile these trends if the propagation time scale for Ronne HSSW through the FRIS cavity is on the order of several months?

We find this consistent with the existence of a direct and a less direct pathway for Ronne HSSW throughout the FRIS cavity, as is now more carefully explained and discussed in the second subsection on of the results, together with an explicit statement that: “Such a delayed response to HSSW inflows along less direct pathways also explains that source water salinities at all moorings under Filchner Ice Shelf continue to rise, when source water salinities at Site 5 gradually decline during 2018/19”.

L193-201: As the authors note, the de-trended, 36-month averaged correlation coefficient is only ~0.5, so only around 25% of the variance in SFR is explained by the SAM. This suggests that there is a connection, but rather a weak one. The authors should be more precise about the strength of this connection in here - saying that they “coincide” overstates that connection, in my opinion.

See also our answer to the general comments. We rephrased the text to clarify this more precisely: “Some clusters of enhanced SFR events (Fig. 1D) occurred during periods of SAM (Methods, Fig. 1C) in a stronger positive phase (Methods, Fig. 1C). In particular the increase since 2015 that caused the reinforcement of the Ronne mode observed at our moorings, as well as the extreme maximum in 1998 that also affected the cavity circulation (Nicholls et al 2001) coincide with periods where the 36 month low-pass filtered SAM index is more positive than a linear regression of the time series.”

Fig. 4: The caption indicates that speed is indicated by contours, but I think the contours indicate pressure and the colors indicate speed.

We corrected that error in the revised version of the manuscript.

L221-222: It is important to clarify here that SFR is defined using the time-mean wind speed. The full SFR, SFR, estimated using the time-varying wind speed is relatively strongly correlated with both the open water area and with the wind speed itself.*

The following explanatory statement has been included in the revised version of the manuscript: “Note that to suppress self-correlations with atmospheric pressure fields in the composite average, SFRs were computed using a spatially varying, but time averaged (1979 to 2019) wind speed, whereas an estimate using the time-varying wind speed (SFR*) is relatively strongly correlated with both the open water area and with the wind speed itself (see methods). However, the composite pattern remains unchanged for

both definitions SFR and SFR*, supporting robustness of the relation between the strength of the southerly winds in the Southern Weddell Sea and activity of the Ronne polynya.”

L228: This is objectively not a strong correlation (10% of variance explained). In general I'm not sure that the evidence presented supports a strong link between the Weddell Sea and the ASL.

We have added a moderating statement when reporting this result: “Monthly time series of the difference in pressure anomaly between these locations significantly, albeit weakly, correlate at zero lag with the SFRs ($r = .33$).” However, as explained in our general response, we did not want to present this as evidence for a link between the Eastern Weddell Sea pressure anomaly and the ASL.

L234-235: Table S2 shows that the correlation between the ASL longitude and the EWS SLP is -0.03, which seems to contradict this claim.

As explained in our general response, we did not intend to establish a link between the EWS SP anomalies and the ASL position, but rather highlight two factors, a positive pressure anomaly in the EBS and a low pressure anomaly in the EWS, that may favor offshore winds over the Ronne Ice Shelf Front, as can be seen from the composite analysis. None of them seem to play a sole dominating role, but the EBS anomalies appear to be linked to the ASL position, while the EWS anomalies may loosely be related to SAM, and both factors appear to have acted in concert during the recent phase of increased dense water production in the southern Weddell Sea. We hope that after our revisions, the reviewer may see the value in putting forward this physically plausible mechanism, although the statistical evidence is too weak for a solid proof.

L254-255: Please clarify this statement. My reading was that Ronne-sourced ISW has not been observed in the Filchner trough previously, which is clearly not correct (e.g. Nicholls et al. 2009).

The sentence was misleading and has been removed in the revised version of the manuscript.

L266-268: Recent changes in the SAM are likely partially due to ozone forcing whereas future changes are predicted to be caused by greenhouse gas forcing (e.g. Shindell and Schmidt 2004, GRL). It is therefore unclear whether continuing increases in the SAM should support continued increases in the katabatic wind strength at the Ronne ice front, particularly if the air masses over the Antarctic continent begin to warm.

See above. We rephrased the text according to the reviewers suggestion in the revised version of the manuscript, in addition to the earlier statement that the SAM influence is not unambiguous (e.g. Dennison, et al., 2014) and less important than the influence of the position and strength of the ASL.

L272: Similar to my comments above, I really think this is overstating the link between the ASL and the dense water formation in the Weddell.

We believe that such a link between different regions around Antarctica may indeed exist, mediated through longitudinal position of cyclones and anticyclones that determines the direction of the atmospheric flow, here in particular the positive pressure anomalies in the eastern Bellingshousen Sea and the adjacent Amundsen Sea Low.

L469-471: In general the community seems to be shifting toward using the TEOS-10 standards i.e. conservative temperature and absolute salinity. I understand that using potential temperature and practical salinity allows more direct comparison with previous measurements, I would recommend that the authors also provide their results (perhaps as a supplement?) in terms of conservative temperature and absolute salinity, to facilitate direct comparison with future studies that use these thermodynamic variables.

We have adapted the TEOS-10 definitions for the primary analysis in the revised version of the manuscript and provide secondary values in units of practical salinity for comparability with previous studies.

L480-482: Do the currents also need to be corrected for tides?

We are not entirely sure what the reviewer is referring to as “correction” in this context, but we may also point to our statements about the analysis of current meter data in our reply to reviewer #2. Usually, tidal a “correction” is employed e.g. when estimating geostrophic (i.e. time mean background) currents from a snapshot observation of velocity (such as typically obtained from a Lowered Acoustic Doppler Current Profiler on a ship), to remove the time-varying tidal component when data is combined from multiple profiles observed at different times. When measuring a velocity time series with a moored current meter, the tidal component is an integral part of the observation that characterizes the flow, and there is no error in the record that would need a correction (such as e.g. for the magnetic declination that is experienced by the compass of the instrument). One may, however, for interpretation of the measurements, separate different time scales of variability, e.g. through low pass filtering that removes most of the oscillatory tidal currents. This is what we have done in our analysis to assess the residual currents associated with the buoyancy driven circulation beneath the ice shelf, and where tidal

dynamics may add another component to the net transport, as is being discussed in the revised version of the manuscript in response to the comments of reviewer #2.

L532-536: I'm not sure I fully understand the rationale here. It is reasonable to ask to what extent the SFR variations are associated with (wind-driven) changes in SIC, and setting U equal to its time mean is a reasonable way of separating these influences. However, I don't understand why SFR, rather than SFR, is used as the primary quantifier of sea ice formation - isn't SFR* the more defensible estimate?*

We agree with the reviewer that SFR* is the more defensible estimate for quantifying the sea ice formation rates, which is why those are used to show time series of absolute daily data in Fig 2D. However, as is now explicitly stated in the methods section in the revised version of the manuscript: "the bulk formulation is basically a linear combination of wind velocity, temperature and sea ice concentration. Hence, if SFR* would be used to select months of the enhanced sea ice formation events for the atmospheric composite analysis, then months with higher wind velocities would a priori be preferably chosen. By using the SFR estimate based on time averaged winds, this autocorrelation is suppressed and the composite provides more robust evidence of a relationship between reduced sea ice concentrations and enhanced off shore winds at the ice shelf front, as two independent variables." The similarity of the composite patterns when using SFR or SFR* (see earlier reply) lend further credibility to the analysis.

L538-547: There are some caveats associated with using reanalysis data around Antarctica that could be more strongly emphasized here. In particular, different products exhibit varying quality of agreement with measurements from meteorological stations (Bracegirdle and Marshall 2012, J. Climate), and varying decadal trends in wind speed and SLP (Hazel and Stewart 2019, J. Climate). In particular, the authors might consider switching to ERA5, rather than ERA-Interim (Dong et al. 2020, J. Climate).

We agree that reanalysis data have their limits, especially around Antarctica due to the low number of observations. Biases in temperature, precipitation and also spurious trends are found in some variables/periods. However, while ERA5 has various improved formulations than its predecessor ERA-Interim, comparisons of both products with observations in the Polar Regions are still few and it remains unclear whether the ERA5 necessarily yields more realistic results than ERA-Interim, which has been used successfully in many studies of the Southern Ocean climate. In our study, the atmospheric reanalysis is foremost used to identify large scale atmospheric patterns that favor wind-driven sea ice formation in front of the Ronne Ice Shelf. As discussed in our reply to the reviewer's comment on the use of SFRs and SFR*s, these periods are primarily identified through anomalous low sea ice concentration diagnosed from the product that is independent of the reanalysis being used. Hence, the selection of months to be included in the composite average of atmospheric anomalies will also primarily be determined by the sea ice product and it is reasonable to assume that most reanalysis products will capture similar large scale circulation anomalies during those times, despite the differences they may display under a more detailed comparison. In fact, using the months of enhanced SFR events shown in fig. 1D to compute an

atmospheric composite from ERA5 monthly anomalies shows a very similar picture than the one obtained from the ERA-Interim data (see below). Hence we believe that ERA interim is fully sufficient for our purpose and the use of ERA5 would not change our results. A discussion of these issues has been added to the methods subsection “Climatic indices, atmospheric composite and significance”, together with the description on the composite averages.

Fig. R1: Comparison of composites of atmospheric anomalies associated with enhanced SFR events identified in Fig. 1D of the main manuscript based on ERA5 (left) and ERA-Interim (right) as is shown in Fig. 5 of the main manuscript.

L610: Spurious period after “<SFR>”

Corrected

Fig. S2 caption: Do you mean “diamonds” rather than “stars”?

Corrected

References:

Daae, K. et al. Necessary Conditions for Warm Inflow Toward the Filchner Ice Shelf, Weddell Sea. *Geophysical Research Letters* 47, e2020GL089237,

Darelius, E., Fer, I. & Nicholls, K. W. Observed vulnerability of Filchner-Ronne Ice Shelf to wind-driven inflow of warm deep water. *Nature Communications* 7, 12300 (2016).

Darelius, E. & Sallée, J. B. Seasonal Outflow of Ice Shelf Water Across the Front of the Filchner Ice Shelf, Weddell Sea, Antarctica. *Geophysical Research Letters* 45, 3577-3585, doi:10.1002/2017GL076320 (2018).

Dennison, F. W., A. J. McDonald, and O. Morgenstern, 2014: The effect of ozone depletion on the Southern Annular Mode and stratosphere-troposphere coupling. *J. Geophys. Res., Atmos.*, 120, 6305-6312, doi:10.1002/2014JD023009.

Hazel, J. E. & Stewart, A. L. Bi-stability of the Filchner-Ronne Ice Shelf Cavity Circulation and Basal Melt. *Journal of Geophysical Research: Oceans* n/a, e2019JC015848, doi:10.1029/2019JC015848 (2020).

Janout, M. et al. FRIS revisited in 2018: On the circulation and water masses at the Filchner and Ronne ice shelves in the southern Weddell Sea. *Journal of Geophysical Research: Oceans* (in revision).

Makinson, K. & Nicholls, K. W. Modeling tidal currents beneath Filchner-Ronne Ice Shelf and on the adjacent continental shelf: Their effect on mixing and transport. *Journal of Geophysical Research: Oceans* 104, 13449-13465,

McKee, D. C., Yuan, X., Gordon, A. L., Huber, B. A., and Z. Dong, 2011: Climate impact on interannual variability of Weddell Sea Bottom Water. *J. Geophys. Res.*, 116, C05020, doi:10.1029/2010JC006494.

Nansen, F., 1902: The oceanography of the North Polar Basin. *Norwegian North Polar Expedition 1893-1896. Scientific Results. Vol. III*, 1-427.

Nicholls, K. W. Predicted reduction in basal melt rates of an Antarctic ice shelf in a warmer climate. *Nature* 388, 460-462, doi:10.1038/41302 (1997).

Nicholls, K. W., Østerhus, S., Makinson, K., Gammelsrød, T. & Fahrbach, E. Ice-ocean processes over the continental shelf of the southern Weddell Sea, Antarctica: A review. *Reviews of Geophysics* 47, doi:10.1029/2007RG000250 (2009).

Nicholls, K. W., Østerhus, S., Makinson, K. & Johnson, M. R. Oceanographic conditions south of Berkner Island, beneath Filchner-Ronne Ice Shelf, Antarctica. *Journal of Geophysical Research: Oceans* 106, 11481-11492, doi:10.1029/2000JC000350 (2001).

Nøst, O. A. & Østerhus, S. Impact of grounded icebergs on the hydrographic conditions near the Filchner Ice Shelf. *Antarctic Research Series* 75, 267-284 (1998).

Silvano, A. et al. Freshening by glacial meltwater enhances melting of ice shelves and reduces formation of Antarctic Bottom Water. *Science Advances* 4, eaap9467, doi:10.1126/sciadv.aap9467 (2018).

Hellmer, H. H., Kauker, F., Timmermann, R. & Hattermann, T. The Fate of the Southern Weddell Sea Continental Shelf in a Warming Climate. *Journal of Climate* 30, 4337-4350, doi:10.1175/JCLI-D-16-0420.1 (2017).

Wadhams, P., 2000. *Ice in the ocean*. Gordon and Breach Science Publishers, Amsterdam, ISBN 90-5699-296-1, 351pp.

Yuan, X. and D. G. Martinson, 2001: The Antarctic Dipole and its Predictability. *Geophys. Res. Lett.*, 28 (18), 3609-3612. <https://doi.org/10.1029/2001GL012969>.

Yuan, X., 2004: ENSO-related impacts on Antarctic sea ice: a synthesis of phenomenon and mechanism. *Antarctic Science* 16 (4), 415-425, doi:10.1017/S0954102004002238.

Yuan, X., Kaplan, M.R., and M.A. Cane, 2018: The Interconnected Global Climate System – A Review of Tropical-Polar Teleconnections. *J. Climate*, 31, 5765-5792, doi:10.1175/JCLI-D-16-0637.1

https://origin.cpc.ncep.noaa.gov/products/analysis_monitoring/ensostuff/ONI_v5.php

(Oceanic Niño Index (ONI, NOAA))

REVIEWERS' COMMENTS

Reviewer #1 (Remarks to the Author):

I found the manuscript strongly improved. It is a very important work that connect large scale climate variability with basal melting of the largest (by volume) ice shelf on Earth. I do recommend publication in Nature Communications. I just have few comments below that require small changes in the wording.

Comments

- Line 80: Hellmer et al (2012) suggest that a redirection of the coastal current (due changes in sea ice upstream) is the first cause of the future increase in FRIS basal melt, then HSSW reduction is a consequence of increased melting. Please clarify this in the text.
- Figure 1: include 2 and 10 cm s-1 near the arrows in the figure legend.
- Line 91: "related".
- Line 132: "dominates".
- Line 166-168: I am not sure to follow. Are there observations of melting from the southern cavity? If not, it is better to avoid mentioning regime shift in the "wide FRIS", but restricting to northern FIS, where there are observations of basal melt.
- Line 267: Fig 1A
- Line 270-272: SAM has some implications on meridional atmospheric flow, and it interacts with ENSO. Therefore SAM can affect the meridional winds in the Weddell Sea. I would add a few sentence here to highlight this.
- Line 329-340: also here assuming that SAM has only impact on zonal winds might not be correct. The pressure release to the north as a mechanism to enhanced sea ice formation on the shelf require some further analysis, which I do not think is necessary. Other studies (e.g. Fogt et al., 2011; Mckee et al 2011) provide some hints on SAM impact of non-annular winds. I would just mentioned these studies and avoid further explanation that would require unnecessary analysis.
- Line 340-346: This text seems more appropriate for the Discussion.
- Line 404: cut "wide".
- Line 402: A recent study in the Ross Sea (Silvano et al., 2020 Nat. Geoscience) provide similar evidence of the impact of SAM and ENSO on sea ice formation and HSSW formation there. Mckee et al 2011 also provide some evidence of the connection between remote forcing (SAM and ENSO) and sea ice/HSSW formation in the Weddell Sea. So I would say that this study provide strong evidence on the connection with the FRIS cavity, and this is the key new result here. Link between climate and sea ice formation on the Antarctic shelf is a recent result, but not new. Few more words, especially about previous work in the Weddell Sea, would be useful here (or before in the discussion).
- Line 524: Figure 5.
- Line 536: Figure 4A?

Reviewer #2 (Remarks to the Author):

I congratulate the authors on this version of the paper. I found it easy to read and I understood a lot more about the processes.

I particularly appreciate circulation pathways on Fig 1a, and an astonishing Fig 4. Your discussion of velocity observations is enough, is well-balanced with discussion of salinity signal, and is honestly more than I have expected to be possible. I take your point about Sweden :)

Small comments:

Fig R1 - Could it be in the supplementary? I definitely gained information from it, but I agree that it's not necessary in the main text

Line 175 - "blue and orange patch" - green and orange?

Reviewer #3 (Remarks to the Author):

Review: 'Observed interannual changes beneath Filchner-Ronne Ice Shelf linked to large scale atmospheric circulation' by Hattermann et al.

The authors have thoroughly revised this manuscript to address my and the other reviewers' comments on the original submission. In particular, they now provide a much more nuanced discussion of the evidence for the link between High Salinity Shelf Water formation and atmospheric anomalies. The title and abstract have also been reframed appropriately. I have just a handful of further minor corrections and suggestions for the authors to consider (see below). I therefore recommend publication, and leave it to the authors to incorporate the comments below as they see fit.

Fig. 4E: In Fig. 2C there is a substantial increase in HSSW source water salinity at FNE1 and FNE2 over this time period, but this does not seem to be reflected in the density time series shown here. Why is this?

L91: "is relates"

L119: "begins decrease"

L165: "increase under enhanced"

L201-207: In Fig. 1 the velocity scale for the lowest velocity measurements is 2cm/s, which corresponds to a much longer advective time scale from the Ronne ice front to east of Berkner island - around 15 months, rather than 3 months.

L224: "the, the"

L321: Formatting issue

L437: "sated"

RESPONSE TO REVIEWERS' COMMENTS

Reviewer #1 (Remarks to the Author):

I found the manuscript strongly improved. It is a very important work that connect large scale climate variability with basal melting of the largest (by volume) ice shelf on Earth. I do recommend publication in Nature Communications. I just have few comments below that require small changes in the wording.

Comments

- Line 80: Hellmer et al (2012) suggest that a redirection of the coastal current (due changes in sea ice upstream) is the first cause of the future increase in FRIS basal melt, then HSSW reduction is a consequence of increased melting. Please clarify this in the text.

It is true that Hellmer et al. (2012) initially hypothesized that the increased access of WDW beneath FRIS would be triggered by a redirection of the coastal currents by upstream sea ice changes. However, more careful analysis of the underlying drivers in the same model runs (Hellmer et al. 2017) and recent, new high resolution model simulations (Daae et al. 2020) have provided a more complicated picture of the mechanisms that may lead to FRIS warming in a future climate, particularly highlighting the role of the density barrier in the Filchner Trough. To reflect these updated insights (that were all brought forward by studies that have overlapping authors with the initial Hellmer et al. 2012 study), we included the Daae et al. (2020) reference to properly frame the background knowledge for our present study and to clarify the reviewer's request.

- Figure 1: include 2 and 10 cm s-1 near the arrows in the figure legend.

We tried different ways of including velocity labels directly with the arrows in the figure, but did not find a satisfying configuration for the figure that is already heavily loaded with detailed information. Since we already use two arrows to visualize the vertical placement of the instruments, we prefer to keep the description of the (site-dependent) velocity scale that is represented by the arrow length in the figure caption.

- Line 91: “related”.

Corrected.

- Line 132:”dominates”.

Corrected.

- Line 166-168: I am not sure to follow. Are there observations of melting from the southern cavity? If not, it is better to avoid mentioning regime shift in the “wide FRIS”, but restricting to northern FIS, where there are observations of basal melt.

The relationship between the Ronne-HSSW inflow intensity and melt rates inside the deeper part of the FRIS cavity was initially established by Nicholls (1997), but has later also been shown by different modelling studies, e.g. Hausmann et al. (2019) and Daae et al. (2020). The underlying mechanism, in which the intensity of buoyancy-driven plumes (Jenkins 1991) control the thermohaline circulation and associated heat flux beneath an ice shelf, has been identified as a basic principle of the mode-1 drive melting that is dominant beneath FRIS (Jackobs et al. 1992). Based on these basic ice shelf cavity dynamics, we here logically infer that an increased dense water inflow (as would be associated with the enhanced sea ice formation that we diagnose and that would cause the intensification of the Ronne mode that we observe at our ice shelf cavity moorings) into the deep cavity, will likely act to increase basal melt rates near the grounding lines (and with subsequent consequences of more outflow of potentially super-cooled ice shelf meltwater with refreezing potential at shallower depths, as is being observed at the northern moorings). Hence, we believe that these implications on the temporal variability of melt rates at different parts of the ice shelf are important insights that can be derived from our observations (in conjunction with existing understanding of the dynamics of the system), which we wish to report, in the hope that they will be verified by future spatially and temporally varying basal melt estimates from remote sensing and ground based radar work (in addition to model results that already have shown this behavior). To address the reviewer's concern, we added a reference to the recent modelling study of Hausmann et al. (2020) to the statement that "melting deeper within the FRIS cavity may be expected to increase under enhanced inflow of Ronne HSSW that brings heat deep into the cavity".

- Line 267: Fig 1A

Corrected.

- Line 270-272: SAM has some implications on meridional atmospheric flow, and it interacts with ENSO. Therefore SAM can affect the meridional winds in the Weddell Sea. I would add a few sentence here to highlight this.

It is correct that there is a relationship between SAM and the meridional atmospheric flow (negative SAM index means increased, positive SAM index decreased meridional flow between high and mid-latitudes (e.g. Tietäväinen and Vihma, 2008). The detailed manifestation of these dynamics depend on the considered latitude range, though. However, this is not relevant for our argumentation here, thus we prefer not to mention it here because it would only interrupt the flow of thoughts and distract from the main points of our analysis.

- Line 329-340: also here assuming that SAM has only impact on zonal winds might not be correct. The pressure release to the north as a mechanism to enhanced sea ice formation on the shelf require some further analysis, which I do not think is necessary. Other studies (e.g. Fogt et al., 2011; Mckee et al 2011) provide some hints on SAM impact of non-annular winds. I would just mentioned these studies and avoid further explanation that would require unnecessary analysis.

See comment above. We do not assume that SAM has only an impact on zonal winds. We thank the reviewer for the suggested references of Fogt et. al. and McKee et al., which we have

included in the revised version of the manuscript, as described further below.

- Line 340-346: This text seems more appropriate for the Discussion.

This is probably a matter of taste. We feel that this summary of recent events that coincided within our study region integrates well into the summary of our atmospheric analysis. The discussion in contrast, takes a more overarching perspective and there is no obvious place in the current text where the introduction of these concrete events would fit in without interrupting the flow.

- Line 404: cut “wide”.

Corrected.

- Line 402: A recent study in the Ross Sea (Silvano et al., 2020 Nat. Geoscience) provide similar evidence of the impact of SAM and ENSO on sea ice formation and HSSW formation there. Mckee et al 2011 also provide some evidence of the connection between remote forcing (SAM and ENSO) and sea ice/HSSW formation in the Weddell Sea. So I would say that this study provide strong evidence on the connection with the FRIS cavity, and this is the key new result here. Link between climate and sea ice formation on the Antarctic shelf is a recent result, but not new. Few more words, especially about previous work in the Weddell Sea, would be useful here (or before in the discussion).

The Ross Sea is on the Pacific side of the Southern Ocean and thus the links with ENSO are more direct than for FRIS on the Atlantic side. McKee et al. (2011) mainly investigate statistical relationships between Weddell Sea Bottom Water temperature and ENSO/ SAM and suggest that “anomalous winds may alter production of dense shelf water by modulating open water area over the shelf. Second, surface winds may alter the volume of dense water exported from the shelf by governing the Weddell Gyre’s cyclonic vigor”.

Our study proves their first hypothesis. (They also hypothesize that the dense water exported from the shelf is influenced by the surface winds, which are related to SAM, but not in a straightforward way, as was also recently explored by Le Paih et al. 2020.)

We included a short discussion of the two suggested references in the discussion, including the references suggested by the reveiwer:

“A very strong warm ENSO event in 2015/16 (defined using the ONI (NOAA)) might be connected to the increase in SFR after 2016 found in this study. Those ENSO-induced anomalies in Antarctica are modulated by SAM depending on their phase relationship^{57,58}. McKee et al. found⁵⁴ statistically significant relationships between Weddell Sea Bottom Water temperature and ENSO/SAM. They suggest that dense shelf water production might be influenced by anomalous winds via modulating the fraction of open water over the shelf. Our study proves this hypothesis and similar mechanisms were found for the Ross Sea in the Pacific sector of the Southern Ocean⁵⁹, which is under more direct influence of ENSO.”

- Line 524: Figure 5.

Corrected.

- Line 536: Figure 4A?

The correct reference is Figure 5a.

Reviewer #2 (Remarks to the Author):

I congratulate the authors on this version of the paper. I found it easy to read and I understood a lot more about the processes.

I particularly appreciate circulation pathways on Fig 1a, and an astonishing Fig 4. Your discussion of velocity observations is enough, is well-balanced with discussion of salinity signal, and is honestly more than I have expected to be possible. I take your point about Sweden :)

Small comments:

Fig R1 - Could it be in the supplementary? I definitely gained information from it, but I agree that it's not necessary in the main text

We thank the reviewer for this comment, which has motivated us to include this figure in the revised version of the companion paper to our study of Janout et al., which presents the results from the open ocean survey along the FRIS-front, and added a reference to it in the main text:

“While being most visible from changes in temperature and salinity, which contain information about the history of the water masses, the current meter records also confirm a shift from a largely partitioned cavity circulation where local inflows dominate the northern Filchner Ice Shelf (Berkner mode, green to purple arrows in Fig. 1a), toward a coherent circulation that spans the entire FRIS in response to increased HSSW inflow at the Ronne ice front (Ronne mode, orange to blue arrows in Fig 1a), as discussed in further detail below and with a schematic of the associated overturning circulation being depicted in Fig. 14 in Janout et al.¹¹.”

Line 175 - “blue and orange patch” - green and orange?

Corrected

Reviewer #3 (Remarks to the Author):

Review: ‘Observed interannual changes beneath Filchner-Ronne Ice Shelf linked to large scale atmospheric circulation’ by Hattermann et al.

The authors have thoroughly revised this manuscript to address my and the other reviewers' comments on the original submission. In particular, they now provide a much more nuanced discussion of the evidence for the link between High Salinity Shelf Water formation and atmospheric anomalies. The title and abstract have also been reframed appropriately. I have just a handful of further minor corrections and suggestions for the authors to consider (see below). I therefore recommend publication, and leave it to the authors to incorporate the comments below as they see fit.

Fig. 4E: In Fig. 2C there is a substantial increase in HSSW source water salinity at FNE1 and FNE2 over this time period, but this does not seem to be reflected in the density time series shown here. Why is this?

This is, because there exists no one-to-one relationship between density and source water salinity, which is why this index is so useful for our analysis. During the transition from the Berkner mode to Ronne mode the density stays relatively constant, because although the Berkner-HSSW is significantly cooler than the Ronne-HSSW derived ISW, both water masses have relatively similar salinity, which at cold temperatures dominates changes in density. In this way, we find that they must relate to different sources, because a cooling through basal melting would always induce an associated freshening (and hence decrease in density) along the meltwater-mixing line that was first derived by Gade (1969). In fact, there is likely a causal relationship due to which the two water masses have similar density, where the outflow of Ronne-HSSW derived ISW forms the precursor water mass for Berkner HSSW formation. This is briefly described in the discussion section:

“Although varying amounts of remotely sourced Ronne ISW and more locally produced Berkner HSSW have been observed in front of Filchner Ice Shelf (as indicated by blue and red letters in Fig. 1d), the density in the lower Filchner Trough has remained remarkably stable throughout recent decades^{11,47}. This suggests that such feedbacks indeed exist, where the outflow of Ronne HSSW-sourced ISW acts as a precursor for the Berkner HSSW production (i.e. by determining the density at which convection that regulates HSSW production may occur)”.

While the subject is dealt with in more detail in the related study of Janout et al. that is currently under review and endorsed with our submission, we added the following (bold) description of the source salinity index in the methods description to clarify the reviewer's concern:

“Based on this, time series of source water salinity were computed from the filtered in-situ temperature and salinity data from the lowermost CTD instrument at each ice shelf mooring (Fig. 2c) that allow to compare the origins of water masses that have experienced different degree of modification due to their interaction the ice shelf.”

L91: “is relates”

Corrected.

L119: “begins decrease”

Corrected

L165: “increase under enhanced”

Corrected.

L201-207: In Fig. 1 the velocity scale for the lowest velocity measurements is 2cm/s, which corresponds to a much longer advective time scale from the Ronne ice front to east of Berkner island - around 15 months, rather than 3 months.

The discussion of the advective time scales was substantially adjusted in the previous revision based on the comments of reviewer #2, who now seems to be more satisfied with our description of the current meter observations. As stated in this discussion, the mean flow velocities at the Filchner mooring sites need to be understood as relatively small residuals of a much larger tidally dominated circulation. Therefore, the time scale estimates that do not include the diffusive effects of the tides should be regarded as upper bounds. Furthermore, while it is true that mean flow velocities are relatively small at the southern Filchner sites (FSW & FSE), much higher flow speeds are observed Site 5 (we refer to 10 to 15 cm/s in the text), indicating faster advection upstream of the southern Filchner moorings sites. To address the reviewer’s concern we added the explicit reference to current velocities at the different sites (bold) back to this sentence of our discussion, after it had been removed from that particular sentence to limit redundancy in the revision of this paragraph.

Considering the observed flow speeds between 10-15 cm s⁻¹ at Site 5 and less than 5 cm s⁻¹ at FSW, this is a plausible time scale for the HSSW signal to be advected along an approximately 800 km long pathway from Ronne Ice Front to FSW, although with some uncertainty about contributions from the tides.

L224: “the, the”

Corrected.

L321: Formatting issue

Corrected.

L437: “sated”

Corrected.

References not included in the manuscript:

Jacobs, S., Helmer, H., Doake, C., Jenkins, A., & Frolich, R. (1992). Melting of ice shelves and the mass balance of Antarctica. *Journal of Glaciology*, 38(130), 375-387.
doi:10.3189/S0022143000002252

Jenkins, A. (1991), A one-dimensional model of ice shelf-ocean interaction, *J. Geophys. Res.*, 96(C11), 20671– 20677, doi:[10.1029/91JC01842](https://doi.org/10.1029/91JC01842).

Le Paih, N., Hattermann, T., Boebel, O., Kanzow, T., Lüpkes, C., Rohardt, G., et al. (2020). Coherent seasonal acceleration of the Weddell Sea boundary current system driven by upstream winds. *Journal of Geophysical Research: Oceans*, 125, e2020JC016316.
<https://doi.org/10.1029/2020JC016316>

Tietäväinen , H. and T. Vihma, 2008: Atmospheric moisture budget over Antarctica and the Southern Ocean based on the ERA-40 reanalysis. *Int. J. Climatol.* 28, 1977-1995. DOI: 10.1002/joc.1684.